# Dynamic pricing and assortment under a contextual MNL demand

**Vineet Goyal**
Columbia University
New York, NY
vgoyal@ieor.columbia.edu

**Noemie Perivier**
Columbia University
New York, NY
np2708@columbia.edu

## Abstract

We consider dynamic multi-product pricing and assortment problems under an unknown demand over T periods, where in each period, the seller decides on the price for each product or the assortment of products to offer to a customer who chooses according to an unknown Multinomial Logit Model (MNL). Such problems arise in many applications, including online retail and advertising. We propose a randomized dynamic pricing policy based on a variant of the Online Newton Step algorithm (ONS) that achieves a $O(d\sqrt{T}\log(T))$ regret guarantee under an adversarial arrival model. We also present a new optimistic algorithm for the adversarial MNL contextual bandits problem, which achieves a better dependency than the state-of-the-art algorithms in a problem-dependent constant $\kappa_2$ (potentially exponentially small). Our regret upper bound scales as $\tilde{O}(d\sqrt{\kappa_2 T} + \log(T)/\kappa_2)$, which gives a stronger bound than the existing $\tilde{O}(d\sqrt{T}/\kappa_2)$ guarantees.

## 1 Introduction

In this paper, we consider the contextual dynamic pricing and assortment optimization problems faced by a seller who sequentially observes a contextual demand under bandit feedback. The goal is to learn the underlying model parameters in order to maximize the seller's profit. These problems arise in numerous applications, including pricing and product recommendation in online retail, as well as click-through rate predictions for web search results. Sequential learning is especially important in settings involving short selling seasons or where no historical data is available.

**Problem formulation.** In both the dynamic pricing and assortment problems, we consider a seller who sells a set of $N$ products over a time horizon of $T$ periods. A new customer arrives in each period $t$ and is offered a set of products or prices. In the pricing problem, the customer arrives with a consideration set and the seller has to decide on the prices of the different products in the consideration set. In the assortment problem, the seller selects the subset of products to offer to the customer. In both cases we assume that the customer's purchase decisions are made according to a multinomial logit model (MNL)([32], [28]) which is widely used in modeling customer preferences. In both problems, the objective is to maximize the seller's expected profit, which is equivalent to minimizing its cumulative expected regret (i.e., the difference between the optimal value and the value obtained by following a given policy).

In this paper, we study a feature-based model, where the product utilities are a function of both products and customers features. In particular, we assume that in each period $t$, the customer's utility for each product $j \in [N]$ can be written under the form $x_{t,j}^\top \theta^*$, where $\theta^*$ is an unknown model parameter and $x_{t,j} \in \mathbb{R}^d$ is a feature vector, which can be adversarially chosen (see Sections 2.1 and 3.1 for exact definitions of the problems). Furthermore, we allow feature-dependent price sensitivities in our pricing setting and suppose that the price sensitivities can be expressed as $x_{t,j}^\top \alpha^*$ for some

36th Conference on Neural Information Processing Systems (NeurIPS 2022).

unknown parameter $\alpha^*$. Typically, the number of products is significantly larger than the number of features ($d << N$). The objective is to learn across the products and obtain a regret upper bound scaling with $d$ instead of $N$.

**Literature review.** Non-feature-based dynamic pricing, where the seller sells identical products to the customers over time, was initially investigated by [22] under various assumptions on the demand curve, and has since been extensively studied (see for instance [8], [14], [7] and [15]). We refer the reader to [13] for an in-depth survey of the area. On the other hand, non-feature-based dynamic assortment optimization was first studied by [9] under the assumption of independent demand for the different products. Dynamic assortment under a MNL choice model has been recently considered (see for instance [36], [37] and [39]). In particular, UCB and Thompson sampling-based policies are proposed in [3] and [4]. These policies achieve optimal regrets of $O(\sqrt{NT})$ in the non-feature based setting.

In this paper, we consider feature-based pricing and assortment problems. Feature-based dynamic pricing has recently received a lot of attention. Most of the existing work study a single product pricing setting under various demand models (linear, binary, generalized linear) and assume that the price sensitivity of each product (i.e., the price coefficient in the demand model) is a known constant (see for instance [19], [20], [5] , [12], [27], [26], [24] and [33]). In the single product setting, the most related work is [6], which is the first to incorporate contextual information about the customers under the form of feature-dependent price sensitivities. The benefit of this assumption is illustrated on real datasets. [6] considers a high dimensional setting with a sparsity assumption under a generalized demand model and propose a policy with near-optimal regret. However, they assume i.i.d. features, whereas we consider a more specific demand model (MNL model), but with adversarial features. In [21], is considered a multiple product version of the problem, with feature-based price sensitivities and under the assumption that the demand follows a MNL model. In this last work, the features are also assumed to be i.i.d. Our multi-product pricing setting directly extends [21], and captures the single product setting with unknown (and feature-based) price sensitivity and adversarial features under a binary demand model with logistic noise, for which, to the best of our knowledge, no algorithm with near-optimal regret is known yet.

The MNL feature-based dynamic assortment problem is a variant of the contextual generalized linear bandits problem (see for instance [17], [25] and [2]) with a more complicated state space, in which multiple arms are pulled at the same time. UCB and TS based policies have recently been proposed for the MNL contextual dynamic assortment problem (see for instance [31], [30] and [11]). However, these work suffer from a dependency in a problem-dependent constant $\kappa_2$ (potentially exponentially small), which captures the 'degree of non-linearity' of the problem.

Another related stream of literature is that of combinatorial bandits (see for instance [10], [23] and [34]), and in particular, top k combinatorial bandits ([35]). In this framework, the agent can pull a subset of arms of cardinality less than k in each round and the total reward obtained is a function of the individual rewards for the arms played. However, the reward obtained in our setting for each individual arm depends on the whole set of arms played in period $t$, whereas the rewards are supposed independent in the aforementioned works.

## 1.1   Our contributions.

In this work, we introduce new algorithms for the dynamic contextual pricing and assortment problems. Our main contribution is the following.

**Dynamic pricing**. We present a dynamic pricing policy for the multi-product MNL model with adversarial contexts and feature-dependent price sensitivities that achieves a $O(d \log(T)\sqrt{T})$ regret bound. This is near-optimal given the $\Omega(\sqrt{T})$ lower bound from [21]. Based on structural properties of the MNL model, and more specifically, on the self-concordant-like property of the MNL log likelihood function, we propose to use a variant of the Online Newton Step method ([18]) to update our estimators of the model parameters. We combine it with a random price shock policy to force exploration.

Closest to our pricing setting is [21], in which the authors design a multi-product dynamic pricing policy under a MNL choice model and feature-dependent price sensitivities. The proposed algorithm achieves a $\tilde{O}(\sqrt{T})$ regret. However, it relies on a bayesian assumption, namely the feature vectors

are drawn i.i.d. from some unknown distribution. Our work considers an adversarial context and uses an ONS method to update the parameters. Note that the connection with the Online Convex Optimization framework has been exploited in the literature (see for example the stochastic online gradient descent in [19] for the single product setting without price sensitivities), however, the problem we consider is more challenging since the presence of feature-dependent price sensitivities implies the existence of *uninformative prices* (i.e. prices $p$ such that no pricing policy can learn the true model parameters when repeatedly pricing at price $p$). Note that all prices are informative in [19]. The link between uninformative prices and the difficulty to design low regret policies was first pointed out by [8], which shows that no algorithm can achieve better regret than $\Omega(\sqrt{T})$ in settings involving such prices. We address this challenge by using appropriate randomized price shocks that force exploration (we note that adding random shocks was first used in the context of dynamic pricing in [29]; however, our work is the first to simultaneously use random price shocks and an ONS based update, and the analysis of our policy differs from the analysis in [29]).

Our results imply a $O(d \log(T))$ regret in the single product setting with adversarial contexts and without price sensitivity considered in [19], under the extra assumption that the noise follows a logistic distribution. This improves over the $O(\sqrt{T})$ regret bound of [19] in this case (note that [19] studies more precisely the effect of drifts in the parameters). Note that in the same setting, but with feature-dependent price coefficients, our results also imply a $\tilde{O}(\sqrt{T})$ regret. We have become aware that a concurrent recent paper [40] also obtained a logarithmic regret for the contextual single product pricing problem without price sensitivity through a variant of the online newton method (for a more general demand model with strictly log-concave noise). We would like to underscore that our theoretical results were obtained independently from this work. Furthermore, the algorithm proposed in [40] uses the exp-concavity parameter in the descent step and the regret upper bound provided scales with this potentially exponentially small constant. By leveraging the self-concordance property of the logistic loss, we design an algorithm which does not use such parameter, which may be better in practical applications. However, our regret upper bound still depends on this constant. We present in Appendix D some numerical comparison between the two algorithms, which confirms that our algorithm achieves a significantly better regret than the one in [40] as the exp-concavity parameter decreases.

In addition to our main contribution, we present a new algorithm for MNL contextual bandits.

**MNL contextual bandits**. We consider a setting with uniform product revenues and propose a new UCB-based algorithm for the MNL assortment problem with adversarial contexts. Our algorithm achieves a regret bound of order $\tilde{O}(dK\sqrt{T})$, which is optimal as a function $T$. One major limitation of the state-of-the-art algorithms for the MNL contextual bandits problem ([31], [30] and [11]) is that the regret bounds scale with a problem dependent constant, which we will refer to as $\kappa_2$ (see Section 3 for a formal definition); $\kappa_2$ quantifies the level of non-linearity of the model and can be exponentially small even for moderate size instances ([16]). Our regret bound can be expressed as $C_1 d \sqrt{\sum_{t=1}^{T} \kappa_{2,t}^*} + C_2 d^2 \log(T)/\kappa_2$ (where $\kappa_{2,t}^* \leq 1$ are problem dependent constants, whose value decrease as the model is further away from the linear model). Therefore, we give a significantly stronger bound than [31], whose regret term of order $\sqrt{T}$ scales with $1/\kappa_2$. Note that our first order term improves for small values of $\{\kappa_{2,t}^*\}$. Moreover, a prior knowledge of the value of $\kappa_2$ is often presupposed in the existing algorithms, which may be a major hindrance to their practical implementation. The quantity $\kappa_2$ appears for example in the design of the exploration bonuses of the UCB-MNL algorithm of [31]. Our algorithm does not rely on an a priori knowledge of $\kappa_2$. Our policy is based on optimistic parameter search instead of exploration bonuses, as used in [31]. The analysis relies on a concentration result on the MLE estimators, which uses a generalization of the Bernstein-like tail inequality for self-normalized vectorial martingales in [2] and [16], and leverages the self-concordant property of the MNL log loss.

We would like to mention that a similar result is achieved by [16] and [2] in the logistic bandits setting. Our results cannot, however, be derived from these two works since the MNL contextual bandits problem cannot be formulated as a generalized bandits problem [11]. In this work, we show that the techniques from [16] and [2] can be efficiently extended to the MNL bandit setting, which involves overcoming a few technical challenges that are specific to the MNL problem.

**Notations.** For any vector $x \in \mathbb{R}^d$ and any positive definite matrix $M \in \mathbb{R}^{d \times d}$, let $||x||_M = \sqrt{x^T M x}$. Also let $B_p^d(0, W)$ be the $d$-dimensional ball of radius W under the norm $\ell^p$. In the special case of the norm $\ell^2$, we will drop the index and refer to the $d$-dimensional ball as $B^d(0, W)$. For two symmetric matrices $A, B \in \mathbb{R}^{d \times d}$, $A \succeq B$ means that $A - B$ is semi-definite positive. For $n \in \mathbb{N}$, we use the notation $[n]$ to denote the set $\{1, ..., n\}$. When it is not clear from the context, we use a bold font to denote vectors.

## 2 Multi-product dynamic pricing

### 2.1 Model setting and preliminaries

We consider a dynamic pricing problem for a seller with N products represented by feature vectors $x_1, x_2, ..., x_N \in \mathbb{R}^d$. In each period $t$, a customer arrives with context $z_t \in \mathbb{R}^d$ and a consideration set $S_t \subseteq [N]$, which can be chosen adversarially. For ease of notation, we consider a more general setting where customer features $z_t$ and consideration set $S_t$ are represented by $|S_t|$ adversarially chosen feature vectors $x_{t,1}, ..., x_{t,|S_t|} \in \mathbb{R}^d$. For all $t \in [T]$, let $k_t = |S_t|$ and let $K \le N$ be an upper bound on $k_t$ for all $t$. We also define $X_t$ as the matrix whose rows are $x_{t,j}$ for $j \in [k_t]$ and we refer to it as the context at time $t$. In each period $t$,

1. The seller observes $x_{t,1}, ..., x_{t,k_t} \in \mathbb{R}^d$ and offers prices $p_{t,j}$ for all $j \in [k_t]$.

2. The customer observes the prices, then purchases one *single* product $j \in [k_t] \cup \{0\}$. The probability to purchase each product $j$ is given by:

$$q_{t,j}((\theta^*, \alpha^*), \vec{p_t}) = \begin{cases} \frac{e^{x_{t,j}^\top \theta^* - x_{t,j}^\top \alpha^* p_{t,j}}}{1 + \sum_{i=1}^{k_t} e^{x_{t,i}^\top \theta^* - x_{t,i}^\top \alpha^* p_{t,i}}} & \text{if } j \ge 1 \\ \frac{1}{1 + \sum_{i=1}^{k_t} e^{x_{t,i}^\top \theta^* - x_{t,i}^\top \alpha^* p_{t,i}}} & \text{if } j = 0 \end{cases}, \tag{1}$$

where $\theta^*, \alpha^* \in \mathbb{R}^d$ are two model parameters, which are unknown to the policy maker. For all $j \in [k_t]$, $x_{t,j}^\top \alpha^*$ represents the price sensitivity of customer $t$ for product $j$. Note that the customer has always the possibility to leave without making any purchase (by selecting product $j = 0$).

3. The policy maker observes only the customer's purchase decision. The binary variable $y_{t,j}$ indicates whether the customer has purchased product $j$ at time $t$.

For brevity, let $\gamma^* = (\theta^*, \alpha^*)$ and $\tilde{x}_{t,j} = [x_{t,j}, -p_{t,j} x_{t,j}]$. We can more simply write the utility $u_{t,j} = x_{t,j}^\top \theta^* - x_{t,j}^\top \alpha^* p_{t,j}$ associated with each product as $u_{t,j} = \tilde{x}_{t,j}^\top \gamma^*$. Also, for each $t$ and for a given estimator $\gamma_t$ of the true parameter at time $t$, we denote by $q_{t,j}(\gamma_t, \vec{p_t})$ the estimated purchase probability for product $j$ (obtained by replacing $\gamma^*$ by $\gamma_t$ in (1)).

We make the following assumptions, which are standard in the dynamic pricing literature:

**Assumption 2.1.** For all t, $j \in [k_t]$, $||x_{t,j}||_2 \le 1$.

**Assumption 2.2.** $||(\theta^*, \alpha^*)||_2 \le W$ for some known constant $W$.

Although the contexts $\{x_{t,j}\}_{t \in [T], j \in [k_t]}$ can be adversarially chosen, we need to slightly restrict the set of feasible contexts to guarantee the positiveness of the price sensitivity for all products. Following [21], we make the following assumption, which implies that the price-sensitivity of each product is not too close to zero. This assumption may be reasonable in practice: when the price of a product goes to infinity, its utility should decrease significantly.

**Assumption 2.3.** For all $t \in T, j \in [k_t]$, $x_{t,j}^\top \alpha^* \ge L$ for some known constant $L$.

**Assumption 2.4.** The upper bound $K$ on the number of products in each set $S_t$ is constant.

**Pricing policy and Benchmark.** We consider non-anticipating pricing policies $\pi$, which depend only on the history up to time $t - 1$, $\mathcal{H}_{t-1} = (X_1, \vec{p_1}, \vec{y_1}, ..., X_{t-1}, \vec{p_{t-1}}, \vec{y_{t-1}})$, and the current context $X_t$. The objective is to design a pricing policy so as to minimize the sellers' cumulative expected regret:

$$R^\pi(T) = \sum_{t=1}^{T} \left[ \sum_{i=1}^{k_t} q_{t,i}(\gamma^*, \vec{p_t^*}) p_{t,i}^* - \mathbb{E}^\pi \left( \sum_{i=1}^{k_t} q_{t,i}(\gamma^*, \vec{p_t}) p_{t,i} \right) \right],$$

where $\vec{p_t^*}$ is the optimal vector of prices in period $t$. The expectation is taken over the random feedback and any source of randomization in the policy.

Let $p_{max} := \frac{1+K\max(W,1)}{L} + \frac{1}{K}$. We show in Lemma **??** that for the algorithm we propose, the prices posted at each time $t$ satisfy $p_{t,j} \in [0, p_{max}]$ for all $j \in [k_t]$. Hence we can consider policies $\pi$ that only post prices in $[0, p_{max}]$.

Finally, we define the following constant, which provides information about how much a feasible demand curve can deviate from the linear model (a smaller $\kappa_1$ implies a larger deviation from the linear model).

$$\kappa_1 = \min_{\gamma \in B^{2d}(0,W), t \geq 1, j \in \{1,\ldots k_t\}, \vec{p} \in [0, p_{\max}]^{k_t}} q_{t,j}(\gamma, \vec{p}) q_{t,0}(\gamma, \vec{p}).$$

Note that our pricing policy does not directly use the value of $\kappa_1$. However, this constant still appears in our regret bound (see Section 2.2 for more discussion).

## 2.2 Dynamic pricing policy

Our algorithm combines the two following ingredients: a variant of the ONS method and random price shocks. In particular, we maintain estimators of the parameters which are updated in each iteration by using a variant of ONS on the log likelihood. In each step, our algorithm selects a myopic vector of prices based on the current estimators and adds random price shocks to force exploration and avoid uninformative prices.

We first give the details on the update of the parameters. Given our estimator $\gamma_t$ of the true parameters at time $t$, we let $\ell_t$ denote the log loss at time $t$:

$$\ell_{t,1}(\gamma_t) = -\sum_{j=0}^{k_t} y_{t,j} \log(q_{t,j}(\gamma_t, \vec{p_t})).$$

For a time-dependent sequence of positive regularizers $\{\lambda_t\}_{t \geq 1}$ (the exact value of the regularizers used in our algorithm is given in Theorem 2.5), the estimator obtained before projection after conducting one step of our descent method is:

$$\hat{\gamma}_t = \gamma_{t-1} - \frac{1}{\mu} H_{t-1}^{-1} \nabla \ell_{t-1}(\gamma_{t-1}),$$

where $\mu = \frac{1}{2(1+(1+p_{\max})2\sqrt{6KW})}$ and $H_t = \sum_{s=1}^{t} \nabla^2 \ell_s(\gamma_s) + \lambda_t I_d$ is the regularized Hessian of the negative log likelihood. Note that for the MNL model, $\ell_{t,1}$ is convex (as can be deduced from Lemma **??** 1.), hence $H_t \succ 0$ for all $t$.

Let $B_t$ denote the set $B^{2d}(0,W) \cap \{(\theta, \alpha) \mid x_{t,j}^\top \alpha \geq L \text{ for } j \in [k_t]\}$. $B_t$ represents the set of parameters which satisfy Assumptions 2.2 and 2.3. We obtain the new estimator $\gamma_t$ by projecting $\hat{\gamma}_t$ on the feasible set of parameters:

$$\gamma_t = \Pi_{B_t}^{H_{t-1}}(\hat{\gamma}_t),$$

where $\Pi_{B_t}^{H_{t-1}}(y) = \arg\min_{x \in B_t}(x-y)^T H_{t-1}(x-y)$ is the projection relatively to the norm induced by $H_{t-1}$. As a result, during all the course of the algorithm, our estimator $\gamma_t$ also satisfies the lower and upper boundedness assumptions.

Finally, the seller chooses a perturbation factor $\delta_t = \frac{1}{Wt^{1/4}}$ and computes, independently for each product $j \in [k_t]$, a random price shock $\Delta p_{t,j}$ which takes value $\delta_t$ with probability $1/2$ and $-\delta_t$ with probability $1/2$. The seller posts the vector of prices $\vec{p_t}$ which is the sum of the random price shocks and the myopic vector of prices $g(X_t \alpha_t, X_t \theta_t)$ (see Appendix **??** for a formal definition). The pseudo code of our dynamic pricing policy is presented in Algorithm 1.

**Running time:** The two main computational steps of Algorithm 1 are calculating the inverse of $H_t$ and projecting the parameter back in the set of feasible parameters relative to the norm $H_t$. The time complexity of the first step is $O(d^3)$, which is reasonable when $d$ is not too large as in the setting we consider. The projection step can also be done efficiently by formulating the problem as a Quadratic Programming problem.

Using an online descent method for the parameters estimation allows us to obtain a low regret algorithm despite the presence of adversarial contexts. We would like to note that based on our current analysis, a simpler online method such as a stochastic online gradient descent (as proposed

---

**Algorithm 1** Online Newton method for multiple product pricing

---

**Input:** Upper bound W on $||(\theta^*, \alpha^*)||_2$, lower bound L, sequence of regularizers $\{\lambda_t\}_{t \geq 1}$.

Initialize $\theta_1, \alpha_1 \in B_1$
**for all** $t \geq 1$ **do**

For $j \in [k_t]$, let $\Delta p_{t,j} = \begin{cases} \frac{1}{Wt^{1/4}} & \text{w/p } \frac{1}{2} \\ \frac{-1}{Wt^{1/4}} & \text{w/p } \frac{1}{2} \end{cases}$

For $j \in [k_t]$, set $p_{t,j} = g(X_t\alpha_t, X_t\theta_t)_j + \Delta p_{t,j}$
Post prices $\vec{p_t}$
Observe feedback $\vec{y_t}$
Set $\gamma_{t+1} = (\theta_{t+1}, \alpha_{t+1})$ as follows: $\gamma_{t+1} = \Pi_{B_{t+1}}^{H_t}\left(\gamma_t - \frac{1}{\mu}H_t^{-1}\nabla\ell_{t,1}(\gamma_t)\right)$

**end for**

---

in [19] for single-product dynamic pricing without price sensitivity) would not allow us to obtain sublinear regret. We would also like to mention that our method is different from the Online Newton Step presented in [18], which is the classic online analogue of the Newton method. The ONS method moves into the direction of $A_t^{-1}\nabla\ell_{t,1}$, where $A_t^{-1}$ is an approximation of the inverse of the Hessian. In our case, we move directly into the direction of the inverse of the Hessian multiplied by $\nabla\ell_{t,1}$, and leverage self-concordant-like properties of the negative log likelihood function of the MNL model to show the convergence of the estimators. This allows us to avoid using the parameter $1/\kappa_1$ (which corresponds to the exp-concavity parameter $\beta$ in the literature) in the descent step, as is done by the ONS algorithm.

We are ready to present our regret bound.

**Theorem 2.5.** *Setting $\lambda_1 = 1$, $\lambda_t = d\log(t)$ for all $t \geq 1$, there is a constant $C$ depending only on $W, L, K$ such that the regret of Algorithm 1 is bounded as:*

$$R^\pi(T) \leq Cd\log(T)\sqrt{T}.$$

The proof of Theorem 2.5 is presented in Appendix **??**. We would like to point out that, even though our algorithm does not use the parameter $1/\kappa_1$, it still appears within the constant $C$.

Under the assumptions of [19] (single product dynamic pricing with adversarial contexts and constant price sensitivity), and assuming that the noise has a logistic distribution, we can show the following regret bound, which contrasts with the $O(\sqrt{T})$ upper bound established in [19].

**Corollary 2.6.** *If $k_t = 1$ for all $t$, and if the price coefficient is a known constant, then setting $\lambda_1 = 1$, $\lambda_t = d\log(t)$ for all $t \geq 1$, and letting $\Delta p_{t,1} = 0$ at each step, there is a constant $C$ depending only on $W$ such that the regret of Algorithm 1 with regularizers $\{\lambda_t\}$ and price shocks $\{\Delta p_{t,1}\}$ is bounded as:*

$$R^\pi(T) \leq Cd\log(T).$$

## 2.3 High level ideas and sketch of the proof.

We provide here the main ideas in the proof of Theorem 2.5. The technical details are presented in Appendix **??**. We first follow the classical regret analysis for dynamic pricing policies and decompose the regret between a term due to the error in the estimation of the parameters and a term due to the random price shocks. In particular, we have the following lemma.

**Lemma 2.7.** *There exist constants $C_1, C_2$ depending only on $W, L, K$, such that:*

$$R^\pi(T) \leq \mathbb{E}\left[C_1 C_2 \sum_{t=1}^{T}\sum_{j=1}^{k_t}[((\alpha_t - \alpha^*)^T x_{t,j})^2 + ((\theta_t - \theta^*)x_{t,j})^2] + C_1\sum_{t=1}^{T}||\Delta\vec{p_t}||^2\right]. \quad (2)$$

Since the variances of the random price shocks are $\frac{1}{W^2\sqrt{t}}$, the second term is $O(\sqrt{T})$. Therefore, to exhibit a $\tilde{O}(\sqrt{T})$ regret bound, it suffices to focus on upper bounding the first term. Note that it follows from (2) that the regret upper bound does not require the global convergence of the estimators

to the true parameters. We only need to show that they converge sufficiently fast in the directions given by the contexts seen throughout the T periods.

For any sequence of prices $\{\vec{p}_t\}_{t=1}^{T}$, our online descent method allows us to control the convergence of the estimated utilities $u_{t,j} = x_{t,j}^{\top}\theta_t - x_{t,j}^{\top}\alpha_t p_{t,j}$ to the true utilities $u_{t,j}^{*} = x_{t,j}^{\top}\theta^{*} - x_{t,j}^{\top}\alpha^{*}p_{t,j}$ for each product $j \in [k_t]$. In particular, we have the following lemma.

**Lemma 2.8.** *There is a constant $\tilde{C}$ depending only on $W, L, K$ such that with probability at least $1 - \frac{\log(T)}{T^2}$,*

$$\sum_{t=1}^{T}\sum_{j=1}^{k_t}(x_{t,j}^{\top}(\theta_t - \theta^{*}) - x_{t,j}^{\top}(\alpha_t - \alpha^{*})p_{t,j})^2 \leq \tilde{C}d\log(T).$$

We present the proof in Appendix **??**. Note that the upper bound in Lemma 2.8 is valid for any sequence of prices posted by the seller. However, it is not possible, in general, to derive directly Theorem 2.5 from Lemma 2.8 without an appropriate price experimentation scheme. Suppose at each step we only post the myopic vector of prices $\vec{p}_t = g(X_t\alpha_t, X_t\theta_t)$. If the prices $\{p_{t,j}\}$ happened to be uninformative (i.e. $x_{t,j}^{\top}(\theta_t - \theta^{*}) - x_{t,j}^{\top}(\alpha_t - \alpha^{*})p_{t,j} = 0$), then the left-hand side in Lemma 2.8 would be zero and the bound provided would not be useful. However, adding random price shocks allows us to deviate from uninformative prices and to derive an upper bound on the first term of (2) based on Lemma 2.8. This concludes the proof of Theorem 2.5. If there is a single product ($k_t = 1$), and there is no price sensitivity (i.e. for all $t$, the coefficient in front of $p_{t,1}$ is a known constant, that we assume to be 1 without loss of generality), note that from Lemma 2.8, we get $\sum_t x_t^T(\theta_t - \theta_*) = O(d\log(T))$. Combining this with (2) and the choice of $\Delta p_{t,1} = 0$ for all $t$, we immediately obtain Corollary 2.6.

In [19], where no price sensitivity is involved and the utility is simply written under the form $u_t = x_t^{\top}\theta^{*}$, the use of a stochastic gradient descent method allows the author to directly obtain a $O(\sqrt{T})$ bound on the sum $\sum_{t=1}^{T} x_t^{\top}(\theta_t - \theta^{*})^2$. Such a bound, when combined with our random price shocks, would only give us a linear regret. By exploiting the special structure of the MNL function and using our variant of the ONS instead of an online gradient descent, we obtain the stronger $O(\log(T))$ bound of Lemma 2.8.

The proof of Lemma 2.8 is based on a lower and an upper bound on $\sum_{t=1}^{T} \ell_{t,1}(\gamma_t) - \ell_{t,1}(\gamma^{*})$. Both involve $\sum_{t=1}^{T}\sum_{j=1}^{k_t}(x_{t,j}^{\top}(\theta_t - \theta^{*}) - x_{t,j}^{\top}(\alpha_t - \alpha^{*})p_{t,j})^2$. The proof of the lower bound exploits the convexity of $\ell_{t,1}$ and is based on a Bernstein inequality for martingales difference sequences. This is similar to the inequality used in [19]. The main technical hindsight lies in the proof of the upper bound. It mimics the analysis of the Online Newton Step method presented in [18], but relies on the specific structure of the gradient and Hessian of $\ell_{t,1}$ for the MNL model. Moreover, it unically exploits the self concordant-like property of $\ell_{t,1}$. Let's first recall the definition of a self-concordant-like function.

**Definition 2.9** (self-concordant-like functions [38]). A convex function $f \in C^3(\mathbb{R}^n)$ is called a self-concordant-like function if:

$$|\phi'''(t)| \leq M_f\phi''(t)||u||_2$$

for $t \in \mathbb{R}$ and $M_f > 0$, where $\phi(t) := f(x + tu)$ for any $x \in \text{dom}(f)$ and $u \in \mathbb{R}^n$.

By adapting the proof of Lemma 4 in [38], we show in Appendix **??** the following property.

**Proposition 2.10.** $\ell_{t,1}$ *is self-concordant-like with* $M_f = (1 + p_{\max})\sqrt{6|S_t|}$.

We also detail in Appendix **??** some useful properties satisfied by self-concordant-like functions.

## 3 Improved algorithm for MNL contextual bandits

### 3.1 Problem formulation

We consider the following MNL dynamic assortment optimization problem, also referred as the MNL contextual bandits. In each period $t$, the seller observes feature vectors $\{x_{t,j}\}_{j=1}^{N} \in \mathbb{R}^d$. As

before, this represents a combination of customer and product features which can be adversarially chosen. The seller needs to decide on the set $S_t \subseteq [N]$ to offer, with $|S_t| \le K$. Given the offered assortment, the customer purchases one *single* product $j \in S_t \cup \{0\}$. Each product $j$ is purchased with probability:

$$q_{t,j}(S_t, \theta^*) = \begin{cases} \frac{e^{x_{t,j}^\top \theta^*}}{1 + \sum_{i \in S_t} e^{x_{t,i}^\top \theta^*}} & \text{if } j \in S_t \\ \frac{1}{1 + \sum_{i \in S_t} e^{x_{t,i}^\top \theta^*}} & \text{if } j = 0 \end{cases} \tag{3}$$

where $\theta^* \in \mathbb{R}^d$ is an underlying model parameter. As before, the binary variable $y_{t,j}$ indicates whether the customer has purchased product $j$ at time $t$. Note that our model encompasses the contextual logistic bandit problem with finitely many arms (which corresponds to the case where $K = 1$). The objective is to minimize the cumulative expected regret over the T periods:

$$R(T) = \sum_{t=1}^T \left[ \sum_{j \in S_t^*} q_{t,j}(S_t^*, \theta^*) - \sum_{j \in S_t} q_{t,j}(S_t, \theta^*) \right],$$

where $S_t^*$ denotes the optimal assortment at time $t$.

In [31], the authors study a more general model where a reward $r_{t,j} \in [0, 1]$ is also revealed for each product at time $t$. We consider the case of uniform rewards ($r_{t,j} = 1$) for all products. Maximizing $\sum_{j \in S_t} q_{t,j}(S_t, \theta)$ over all sets $S \subseteq [N]$ of cardinality at most K is now equivalent to selecting the K products which have the highest utility $x_{t,j}^\top \theta$. Hence the set $S_t$ as well as the optimal set $S_t^*$ always contain exactly K products.

Similarly as in the pricing setting, we make the following two assumptions:

**Assumption 3.1.** For all $t \in [T]$, $j \in [N]$, $\|x_{t,j}\|_2 \le 1$.

**Assumption 3.2.** $\|\theta^*\|_2 \le W$ for some known constant W.

Following [31], we also introduce the following constant, which typically appears in connection to the link function in the generalized linear bandits ([17], [25]).

$$\kappa_2 := \min_{|S| \le K, j \in S, t \ge 1} \min_{\|\theta^*\| \le W} q_{t,j}(S, \theta) q_{t,0}(S, \theta) > 0.$$

A smaller value of $\kappa_2$ can be interpreted as a bigger deviation from the linear model. As mentioned before, the regret bound of the dynamic assortment policies of [31] and [11] exhibit a harmful dependency in $\kappa_2$. Besides, an a priori knowledge of the value of $\kappa_2$ is presupposed. Our goal is to design a dynamic assortment policy which does not require prior knowledge of $\kappa_2$ and achieves a regret with a better dependency in $\kappa_2$. For all $t \in [T]$, let $\kappa_{2,t}^* = \sum_{j \in S_t^*} q_{t,j}(S_t^*, \theta^*) q_{t,0}(S_t^*, \theta^*)$. $\kappa_{2,t}^*$ represents the degree of non-linearity for the optimal set $S_t^*$ and depends on the unknown parameter $\theta^*$ as well as on the feature vectors present at time $t$. We show that the $\tilde{O}(\sqrt{T})$ term of our regret bound can be replaced by a $\tilde{O}\left(\sqrt{\sum_{t=1}^T \kappa_{2,t}^*}\right)$ term. Note that we always have $\kappa_{2,t}^* \le 1$ hence this is a strict improvement. As a result, a high level of non-linearity at time $t$ induces a smaller $\kappa_{2,t}^*$ and positively impacts the regret.

### 3.2 Dynamic assortment policy

We design a tight confidence set for the true parameter $\theta^*$ and use it to construct upper confidence bounds on the utility of each product at time $t$. Our algorithm relies on optimistic parameter search over the confidence interval, as used by [2] in the logistic bandits setting. However, in our setting, the seller's decision at time $t$ involves the choice of multiple products. Hence we cannot build a unique optimistic estimator $\theta_t$ as in [2]. The key idea is to do the optimistic parameter search independently for each product, generating a set of parameters $\{\tilde{\theta}_{t,j}\}_{j=1}^N$ such that with high probability, $x_{t,j}^\top \tilde{\theta}_{t,j}$ is an upper bound on the utility $x_{t,j}^\top \theta^*$ of product $j$.

**Confidence set.** The main ingredient is the design of a confidence set for $\theta^*$. We classically start by computing the maximum likelihood estimator of $\theta^*$. Let $\hat{\theta}_t$ be the unique minimizer of the

following function, for a sequence of time-dependent regularizers $\{\lambda_t\}_{t=1}^T$ (the exact values are given in Theorem 3.4):

$$\mathcal{L}_t^{\lambda_t}(\theta) = -\sum_{s=1}^{t-1}\sum_{j\in S_s} y_{s,j}\log(q_{s,j}(S_s,\theta)) + \tfrac{\lambda_t}{2}\|\theta\|^2.$$

$\hat{\theta}_t$ satisfies the equation $\nabla\mathcal{L}_t^{\lambda_t}(\theta) = 0$, where the gradient of $\mathcal{L}_t^{\lambda_t}(\theta)$ is given by: $\nabla\mathcal{L}_t^{\lambda_t}(\theta) = \sum_{s=1}^{t-1}\sum_{j\in S_s}(q_{s,j}(S_s,\theta) - y_{s,j})x_{s,j} + \lambda_t\hat{\theta}_t$. Following [2], for all $\delta \in [0,1)$, we now define a confidence set $C_t(\delta)$ for $\theta^*$ as follows:

$$C_t(\delta) := \{\theta \in \Theta | \|g_t(\theta) - g_t(\hat{\theta}_t)\|_{H_t^{-1}(\theta)} \le \gamma_t(\delta)\},$$

where $g_t(\theta) := \sum_{s=1}^{t-1}\sum_{j\in S_s} q_{s,j}(S_s,\theta)x_{s,j} + \lambda_t\theta$, where $H_t(\theta)$ is the Hessian of the regularized negative log likelihood evaluated at $\theta$, i.e.,

$$H_t(\theta) := \sum_{s=1}^{t-1}\left[\sum_{i\in S_s} q_{s,i}(S_s,\theta)x_{s,i}x_{s,i}^T - \sum_{i\in S_s}\sum_{j\in S_s} q_{s,i}(S_s,\theta)q_{s,j}(S_s,\theta)x_{s,i}x_{s,j}^T\right] + \lambda_t I_d$$

and where

$$\gamma_t(\delta) = \sqrt{\lambda_t}(W + \tfrac{1}{2}) + \frac{2d}{\sqrt{\lambda_t}}\log\left(\frac{4}{\delta}\left(1 + \frac{2tK}{d\lambda_t}\right)\right).$$

The following proposition is the analogue, in the multi-product setting, of Proposition 1 in [2] and establishes that $C_t(\delta)$ is a confidence set for $\theta^*$. The details are given in Appendix **??**.

**Proposition 3.3.** *Let $\delta \in (0,1]$. Then $\mathbb{P}(\forall t, \theta^* \in C_t(\delta)) \ge 1 - \delta$.*

The proof of Proposition 3.3 builds upon the new Bernstein-like tail inequality for self-normalized vectorial martingales presented in [16]. However, Theorem 1 in [16] does not directly apply to our setting because of the correlation between the variables $\{\varepsilon_{t,j} := y_{t,j} - q_{t,j}(S_t,\theta^*)\}_{j\in S_t}$ induced by the presence of multiple purchase options in period $t$. We thus present a generalization of Theorem 1 in [16] for the multiple products setting that handles such correlation.

**Algorithm.** Before describing our algorithm, let's introduce the following notation, which generalizes the choice probabilities given by (3) to the case where the model parameters corresponding to each product are uncorrelated. More precisely, we now consider some estimator $\tilde{\theta} \in \mathbb{R}^{d\times N}$ of the true parameters. For all $j \in [N]$, $\tilde{\theta}_j$ represents the parameter associated with product $j$. The estimated probability that item $j$ is purchased at time $t$ if assortment $S \subseteq [N]$ is offered is computed as:

$$\tilde{q}_{t,j}(S,\tilde{\theta}) = \begin{cases} \frac{e^{x_{t,j}^\top\tilde{\theta}_j}}{1+\sum_{i\in S} e^{x_{t,i}^\top\tilde{\theta}_i}} & \text{if } j \in S \\ \frac{1}{1+\sum_{i\in S} e^{x_{t,i}^\top\tilde{\theta}_i}} & \text{if } j = 0 \end{cases}$$

Now, at time $t$, our algorithm uses the previous contexts and observations to compute the maximum likelihood estimator $\hat{\theta}_t$ as defined above and constructs the confidence set $C_t(\delta)$. Then, for each product $j \in [N]$, the algorithm finds an optimistic parameter $\tilde{\theta}_{t,j} = \arg\max_{\theta\in C_t(\delta)} x_{t,j}^\top\theta$. We offer the set $S_t$ of the K items maximizing the optimistic expected revenue $\tilde{r}_t(S,\tilde{\theta})$:

$$\tilde{r}_t(S,\tilde{\theta}) = \sum_{j\in S}\tilde{q}_{t,j}(S,\tilde{\theta}).$$

Since we assumed all prices to be unit, this is equivalent to offering the $K$ products with highest $x_{t,j}^\top\tilde{\theta}_{t,j}$. In the case of non-uniform revenues, our algorithm is still valid; however, an extra factor $1/\kappa_2$ would appear in the regret upper bound with our current analysis. We also note that Algorithm 2 is mainly of theoretical interest, since computing each $\tilde{\theta}_{t,j}$ remains computationally expensive.

We now present the our upper bound on the regret of our policy.

**Theorem 3.4.** *For $\delta = \frac{1}{K^2T^2}$, $\lambda_1 = 1$, and $\lambda_t = d\log(tK)$ for all $t \ge 1$, the regret of Algorithm 2 satisfies, for some constants $\tilde{C}_1$ and $\tilde{C}_2$ that do not depend on $d, K, T$ and that depend only polynomially on $W$:*

$$R(T) \le \tilde{C}_1 Kd\log(KT)\left(\sqrt{\sum_{t=1}^T \kappa_{2,t}^*}\right) + \frac{\tilde{C}_2 d^2 K^4}{\kappa_2}\log(KT)^2.$$

---

**Algorithm 2** OFU-MNL

---

**Input:** Upper bound K on the size of an assortment, $\delta$, sequence $\{\lambda_t\}_{t=1}^T$

> **for all** $t \geq 1$ **do**
>> Observe feature vectors $\{x_{t,1}, ..., x_{t,N}\}$
>> Set $\hat{\theta}_t = \arg\min \mathcal{L}_t^{\lambda_t}(\theta)$
>> For $j = 1, ..., N$, set $\tilde{\theta}_{t,j} = \arg\max_{\theta \in C_t(\delta)} x_{t,j}^\top \theta$
>> Offer set $S_t = \arg\max_{|S| \leq K} \tilde{r}_t(S, \tilde{\theta})$.
>> Receive feedback $\vec{y_t}$
> **end for**

---

### 3.3 High level ideas and sketch of the proof

We first condition on the event that $\theta^* \in C_t(\delta)$, for all $t \geq 1$. Proposition 3.3 shows that this happens with high probability. From Lemma **??** (Appendix **??**), we have the following concentration result on the optimistic parameter $\tilde{\theta}_{t,j}$ associated with each product $j \in [N]$:

$$\|\theta^* - \tilde{\theta}_{t,j}\|_{H_t(\theta^*)} \leq (1 + \sqrt{6KW})\gamma_t(\delta). \tag{4}$$

Now, using the optimistic choice of assortment made by the algorithm as well as the fact that $x_{t,j}^\top \tilde{\theta}_{t,j}$ is an upper bound on the true utility for each product, we can first bound the regret as follows:

$$R(T) = \sum_{t=1}^T \left[ \sum_{j \in S_t^*} q_{t,j}(S_t^*, \theta^*) - \sum_{j \in S_t} q_{t,j}(S_t, \theta^*) \right] \leq \sum_{t=1}^T \left[ \sum_{j \in S_t} \tilde{q}_{t,j}(S_t, \tilde{\theta}_t) - \sum_{j \in S_t} \tilde{q}_{t,j}(S_t, \tilde{\theta}^*) \right]$$

$$\leq \sum_{t=1}^T \sum_{j \in S_t} q_{t,j}(S_t, \theta^*) q_{t,0}(S_t, \theta^*) x_{t,j}^\top (\tilde{\theta}_{t,j} - \theta^*) + R_2(T) := R_1(T) + R_2(T).$$

where $R_2(T)$ is a second order term which we will prove to be of order $O(d^2 K^4 \log(KT)^2/\kappa_2)$. To show that the first term is of order $O\left(Kd\log(KT)\sqrt{\sum_{t=1}^T \kappa_{2,t}^*}\right)$, we first use the concentration result stated in (4), which implies the following upper bound, for some $C > 0$:

$$R_1(T) \leq C\sqrt{d\log(KT)} \sum_{t=1}^T \sum_{j \in S_t} \left[ q_{t,j}(S_t, \theta^*) q_{t,0}(S_t, \theta^*) \|x_{t,j}\|_{H_t(\theta^*)^{-1}} \right].$$

Using that $H_t(\theta^*) \succeq \kappa_2 \sum_{s=1}^T \sum_{j \in S_s} x_{s,j} x_{s,j}^\top$, we could already show that this term is of order $\tilde{O}(\sqrt{T})$ by applying the Elliptical Potential Lemma from [1]. However, we would then obtain a term with linear dependency in $1/\kappa_2$. We show that the local information given by the terms $\{q_{t,j}(S_t, \theta^*) q_{t,0}(S_t, \theta^*)\}$ and the self-concordance-like property of the log loss can be used to derive a tighter bound on the above sum. The complete version of the proof is provided in Appendix **??**.

## 4 Conclusion

In this paper, we study contextual dynamic pricing and assortment optimization problems under a MNL choice model. We present a dynamic pricing policy based on a variant of the Online Newton Step method combined with random price shocks that achieves near-optimal regret for the MNL model with adversarial contexts and feature-dependent price sensitivities. We also propose a new optimistic algorithm for the adversarial MNL contextual bandits problem. Both our algorithms leverage the self-concordant property of the MNL log likelihood function to achieve better dependency on potentially exponentially small parameters than existing algorithms. An interesting research direction would be to extend our results to other choice models, such as the nested logit model, which is another widely used model in the Revenue Management literature.

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
