# A    Self-concordant properties

In this section, we derive some useful properties of self-concordant-like functions which will be used in the subsequent regret proofs. We first remind the reader of the definition of self-concordant-like functions.

**Definition A.1** (self-concordant-like functions [40]). A convex function $f \in C^3(\mathbb{R}^n)$ is called a self-concordant-like function with constant $M_f$ if:

$$|\phi'''(t)| \leq M_f \phi''(t) ||u||_2$$

for $t \in \mathbb{R}$ and $M_f > 0$, where $\phi(t) := f(x + tu)$ for any $x \in \mathbb{R}^n$ and $u \in \mathbb{R}^n$.

Our results essentially rely on the following property of self-concordant-like functions.

**Proposition A.2** (Theorem 4 in [40]). *Let $f : \mathbb{R}^n \to \mathbb{R}$ be a $M_f$- self-concordant-like function and let $x, y \in dom(f)$, then:*

$$e^{-M_f ||y-x||_2} \nabla^2 f(x) \preceq \nabla^2 f(y)$$

Now, for any $X > 0, m \geq 0, d' > 0, \{z_j\}_{j \in [m]} \in \mathbb{R}^{d'}$ satisfying $||z_j||_2 \leq X$ for all $j \in [m]$, and $\{y_j\}_{j \in [m]} \in \mathbb{R}$, we consider the function $\ell : B^{d'}(0, W) \longrightarrow \mathbb{R}$ defined as:

$$\ell(\beta) = -\sum_{i=1}^m y_i z_i^T \beta + \log \left( 1 + \sum_{j=1}^m e^{z_j^T \beta} \right). \tag{5}$$

Note that the negative log likelihood $\ell_{t,1}$ and $\ell_{t,2}$ can be written as

$$
\begin{aligned}
\ell_{t,1}(\gamma) &= -\sum_{i=0}^{|S_t|} y_{t,i} \log(q_{t,i}(\gamma, \vec{p_t})) \\
&= -\sum_{i=1}^{|S_t|} y_{t,i} \tilde{x}_{t,i}^\top \gamma + \left( \sum_{i=0}^{|S_t|} y_{t,i} \right) \log \left( 1 + \sum_{j=0}^{|S_t|} e^{\tilde{x}_{t,j}^\top \gamma} \right) \\
&= -\sum_{i=1}^{|S_t|} y_{t,i} \tilde{x}_{t,i}^\top \gamma + \log \left( 1 + \sum_{j=1}^{|S_t|} e^{\tilde{x}_{t,j}^\top \gamma} \right), \qquad \text{(since } \sum_{i=0}^{|S_t|} y_{t,i} = 1)
\end{aligned}
$$

with $\|\tilde{x}_{t,j}\|_2 = \|[x_{t,j}, -p_{t,j} x_{t,j}]\|_2 \leq \sqrt{1 + p_{max}^2} \|x_{t,j}\|_2 \leq 1 + p_{max}$.
Similarly,

$$\ell_{t,2}(\theta) = -\sum_{i \in S_t} y_{t,i} \log(q_{t,i}(S_t, \theta)) = -\sum_{i \in S_t} y_{t,i} x_{t,i}^\top \theta + \log \left( 1 + \sum_{i \in S_t} e^{x_{t,i}^\top \theta} \right),$$

with $\|x_{t,j}\|_2 \leq 1$.

Hence for all $t \geq 0$, $\ell_{t,1}$ and $\ell_{t,2}$ are of the form given in 5 with $(z_j = \tilde{x}_{t,j}$ and $X = 1 + p_{max})$ and $(z_j = x_{t,j}$ and $X = 1)$, respectively. In particular, $\ell_{t,1}, \ell_{t,2}$ satisfy all properties stated below with the corresponding constants.

**Proposition A.3.** *The function $\ell$ is self-concordant-like with $M_\ell = X\sqrt{6m}$.*

*Proof.* By following the proof of Lemma 4 in [40], we obtain that the for all $a, \mu \in \mathbb{R}^n$, the function $\psi(s) := \log \left( \sum_{i=0}^n e^{a_i s + \mu_i} \right)$ satisfies the inequality:

$$|\psi'''(s)| \leq \sqrt{6} ||a||_2 \psi''(s). \tag{6}$$

Now, let $f(\beta) := \log \left( 1 + \sum_{j=1}^m e^{z_{t,j}^T \beta} \right)$.
Let $d \in \mathbb{R}^d$ and let $\phi(s) := f(\beta + sd) = \log \left( \sum_{i=0}^m e^{a_i s + \mu_i} \right)$, where $\mu_i = \beta^T z_i$, $a_i = d^T z_i$ and

$\mu_0, a_0 = 0$. Then using (6), we obtain:

$$|\phi'''(s)| \leq \sqrt{6}\|a\|_2 \phi''(s) = \sqrt{6}\sqrt{\sum_{i=1}^{m}(d^T z_i)^2}\phi''(s)$$

$$\leq \sqrt{6}\sqrt{\sum_{i=1}^{m}\|d\|_2^2\|z_i\|_2^2}\phi''(s) \leq X\sqrt{6m}\|d\|_2\phi''(s),$$

where the second inequality comes from Cauchy-Schwartz and the last one is since $||z_j||_2 \leq X$ for all $j \in [m]$. This shows that $f$ is self-concordant-like with constant $M_f = X\sqrt{6m}$.

Since $\ell$ is the sum of f and a linear operator (for which the third derivatives are zero), we obtain that $\ell$ is self-concordant-like with constant $X\sqrt{6m}$.

$\square$

**Proposition A.4.** *The hessian of $\ell$ satisfies, for all $u \in \mathbb{R}^d, \beta_1, \beta_2 \in B^{d'}(0,W)$:*

$$u^T\left(\int_0^1\int_0^1 z\nabla^2\ell(\beta_1 + zw(\beta_2 - \beta_1))dzdw\right)u \geq \frac{1}{2(1+X2\sqrt{6m}W)}u^T\nabla^2\ell(\beta_1)u$$

*Proof.* From Proposition A.2, we obtain:

$$u^T\int_0^1\int_0^1 z\nabla^2\ell(\beta_1 + zw(\beta_2 - \beta_1))dzdwu \geq u^T\nabla^2\ell(\beta_1)u\int_0^1\int_0^1 e^{-M_\ell||zw(\beta_2-\beta_1)||_2}zdzdw$$

Besides,

$$\int_0^1\int_0^1 e^{-M_\ell||zw(\beta_2-\beta_1)||_2}zdzdw = \int_0^1 z\left(\frac{1 - e^{-M_\ell z||\beta_1-\beta_2||_2}}{M_\ell z||\beta_1 - \beta_2||_2}\right)dz$$

Noting that $\frac{1-e^{-x}}{x} \geq \frac{1}{1+x}$ for $x > 0$ (derived from $e^x \geq (1+x)$) we obtain:

$$\int_0^1\int_0^1 e^{-M_\ell||zw(\beta_2-\beta_1)||_2}zdzdw \geq \int_0^1 z\left(\frac{1}{1+M_\ell z||\beta_1-\beta_2||_2}\right)dz$$

$$\geq \int_0^1 z\left(\frac{1}{1+M_\ell||\beta_1-\beta_2||_2}\right)dz$$

$$= \frac{1}{2(1+M_\ell||\beta_1-\beta_2||_2)}$$

$$\geq \frac{1}{2(1+X2\sqrt{6m}W)}$$

where the last inequality comes from the fact that $\beta_1, \beta_2 \in B^{d'}(0,W)$.

$\square$

**Proposition A.5.** *The hessian of $\ell$ satisfies, for all $u \in \mathbb{R}^d, \beta_1, \beta_2 \in B^2(0,W)$:*

$$u^T\int_0^1 \nabla^2\ell_t(\beta_1 + z(\beta_2 - \beta_1))dzu \geq \frac{1}{(1+X2\sqrt{6m}W)}u^T\nabla^2\ell_t(\beta_1)u$$

*Proof.* From Proposition A.2 and using the inequality $\frac{1-e^{-x}}{x} \geq \frac{1}{1+x}$ for $x \geq 0$, we obtain:

$$u^T\int_0^1 \nabla^2\ell_t(\beta_1 + z(\beta_2 - \beta_1))dzu \geq u^T\nabla^2\ell(\beta_1)u\int_0^1 e^{-M_\ell||z(\beta_2-\beta_1)||_2}dz$$

$$\geq u^T\nabla^2\ell(\beta_1)u\left(\frac{1 - e^{-M_\ell||\beta_1-\beta_2||_2}}{M_\ell||\beta_1 - \beta_2||_2}\right)$$

$$\geq u^T\nabla^2\ell(\beta_1)u\left(\frac{1}{1+M_\ell||\beta_1-\beta_2||_2}\right)$$

$$\geq u^T\nabla^2\ell(\beta_1)u\frac{1}{(1+X2\sqrt{6m}W)}$$

where the last inequality comes from the fact that $\beta_1, \beta_2 \in B^{d'}(0,W)$.

$\square$

## B  Dynamic pricing technical proofs

### B.1  Proof of Theorem 2.5

We provide here the full proof of the regret upper bound given in Theorem 2.5. We first state a few technical lemmas.

To begin with, we characterize the greedy price corresponding to the current context and some given estimators of the parameters. The proposition below can be found in [23].

**Proposition B.1.** *([23] Proposition 3.1) If the true utility model parameters are $\theta, \alpha$, then the optimal prices $\{p_{t,i}\}_{i \in [k_t]}$ are as follows. For all $t \geq 1$ and product $i \in [k_t]$,*

$$p_{t,i}^* = \frac{1}{x_{t,i}^\top \alpha} + B_t^0 \equiv g(X_t \alpha, X_t \theta)_i \,, \tag{7}$$

*where $B_t^0$ is the unique fixed point $B$ of the following equation:*

$$B = \sum_{i=1}^{k_t} \frac{1}{x_{t,i}^\top \alpha} e^{-(1+x_{t,i}^\top \alpha B)} e^{x_{t,i}^\top \theta} \,. \tag{8}$$

The following lemma then guarantees the boundedness of the prices posted by Algorithm 1 as well as the boundedness of the resulting product utilities. The proof is deferred to Appendix B.2

**Lemma B.2.** *Let $\vec{p_t} = g(X_t \alpha_t, X_t \theta_t) + \vec{\Delta p_t}$ be the vector of prices posted at time $t$ and let $p_{max} := \frac{1 + K \max(W,1)}{L} + \frac{1}{W}$. Then, for all $j \in [k_t]$, we have $g(X_t \alpha_t, X_t \theta_t)_j \in [0, p_{max}]$ and $p_{t,j} \in [0, p_{max}]$.*

*It follows that the product utilities satisfy $\tilde{x}_{t,j}^\top \gamma_t \leq W(1 + p_{max}) \equiv M$ for all $t \geq 1, j \in [k_t]$.*

We now borrow two lemmas from [23]. First, let $h_t(\vec{p}) = \sum_{j=1}^{k_t} q_{t,j}(\gamma^*, \vec{p}) p_j$ be the expected revenue at time t after posting prices $\vec{p}$. The following lemma shows the boundedness of $||\nabla^2 h_t(\vec{p})||_2$.

**Lemma B.3.** *([23], proof of Lemma 5.4) There is a constant $C_1$ depending only on W, L and K such that the operator norm $||\nabla^2 h_t(\vec{p})||_2$ satisfies $||\nabla^2 h_t(\vec{p})||_2 \leq C_1$ for all $t \geq 0$ and $\vec{p} \in [0, p_{\max}]^{k_t}$.*

The next lemma gives an upper bound in the difference of myopic prices for different values of the estimators.

**Lemma B.4** ([23], Lemma 5.2). *Let $\gamma_1 = (\theta_1, \alpha_1) \in B^{2d}(0, W)$ and $\gamma_2 = (\theta_2, \alpha_2) \in B^{2d}(0, W)$ be such that for all $j \in [k_t]$, $x_{t,j}^\top \alpha_i \geq L$. Then there exists a constant $C_2$ depending only on $W, L, K$ such that:*

$$||g(X_t \alpha_1, X_t \theta_1) - g(X_t \alpha_2, X_t \theta_2)||_2^2 \leq C_2 (||X_t(\alpha_t - \alpha^*)||_2^2 + ||X_t(\theta_t - \theta^*)||_2^2)$$

We are now ready to present the proof of Theorem 2.5.

**Proof of Theorem 2.5.**

Recall that $h_t(\vec{p}) = \sum_{j=1}^{k_t} q_{t,j}(\gamma^*, \vec{p}) p_j$ is the expected revenue at time t after posting prices $\vec{p}$. Since for all $t \geq 0$, $\vec{p_t^*} = \arg \max h_t(\vec{p})$, we have that $\nabla h_t(\vec{p_t^*}) = 0$.

Hence, by doing a Taylor expansion of $h_t$ at $\vec{p_t^*}$, we obtain that the regret is bounded as follows:

$R^\pi(T)$

$$= \mathbb{E}\left[ \sum_{t=1}^{T} \sum_{j=1}^{k_t} q_{t,j}(\gamma^*, \vec{p_t^*}) p_{t,j}^* - q_{t,j}(\gamma^*, \vec{p_t}) p_{t,j} \right]$$

$$= \mathbb{E}\left[ \sum_{t=1}^{T} h(\vec{p_t^*}) - h(\vec{p_t}) \right]$$

$$= \mathbb{E}\left[\sum_{t=1}^{T}(\nabla h(\vec{p_t^*})(\vec{p_t} - \vec{p_t^*}) + \frac{1}{2}(\vec{p_t} - \vec{p_t^*})^\top \nabla^2 h(\vec{\tilde{p}})(\vec{p_t} - \vec{p_t^*}))\right]$$

$$= \mathbb{E}\left[\sum_{t=1}^{T}\frac{1}{2}(\vec{p_t} - \vec{p_t^*})^\top \nabla^2 h(\vec{\tilde{p}})(\vec{p_t} - \vec{p_t^*})\right] \qquad\qquad (\nabla h_t(\vec{p_t^*}) = 0)$$

$$\leq \mathbb{E}\left[\frac{C_1}{2}\sum_{t=1}^{T}||\vec{p_t} - \vec{p_t^*}||_2^2\right]. \qquad\qquad\qquad\qquad (\text{Lemma B.3})$$

$$= \mathbb{E}\left[\frac{C_1}{2}\sum_{t=1}^{T}||g(X_t\alpha_t, X_t\theta_t) + \vec{\Delta p_t} - g(X_t\alpha^*, X_t\theta^*)||_2^2\right]$$

$$\leq \mathbb{E}\left[\frac{C_1}{2}\sum_{t=1}^{T}||g(X_t\alpha_t, X_t\theta_t) - g(X_t\alpha^*, X_t\theta^*)||_2^2 + \frac{C_1}{2}\sum_{t=1}^{T}||\vec{\Delta p_t}||_2^2\right]$$

$$\leq \mathbb{E}\left[\frac{C_1 C_2}{2}\sum_{t=1}^{T}\sum_{j=1}^{k_t}((\alpha_t - \alpha^*)^T x_{t,j})^2 + ((\theta_t - \theta^*)^T x_{t,j})^2\right] + \mathbb{E}\left[\frac{C_1}{2}\sum_{t=1}^{T}||\vec{\Delta p_t}||_2^2\right] \quad (\text{Lemma B.4})$$

$$\leq \mathbb{E}\left[\frac{C_1 C_2}{2}\sum_{t=1}^{T}\sum_{j=1}^{k_t}((\alpha_t - \alpha^*)^T x_{t,j})^2 + ((\theta_t - \theta^*)^T x_{t,j})^2\right] + C_1\sum_{t=1}^{T}\frac{K}{W^2\sqrt{t}}$$

$$\leq \mathbb{E}\left[\frac{C_1 C_2}{2}\sum_{t=1}^{T}\sum_{j=1}^{k_t}((\alpha_t - \alpha^*)^T x_{t,j})^2 + ((\theta_t - \theta^*)^T x_{t,j})^2\right] + \frac{2C_1 K\sqrt{T}}{W^2}, \qquad (9)$$

where in the third equality, $\vec{\tilde{p}}$ is a linear combination of $\vec{p}$ and $\vec{p}^*$.

Using Lemma 2.8, we obtain that with probability at least $1 - \frac{\log_2(T)}{T^2}$:

$$\sum_{t=1}^{T}\sum_{j=1}^{k_t}((\theta_t - \theta^*)^T x_{t,j} - (\alpha_t - \alpha^*)^T x_{t,j}p_{t,j})^2 \leq \tilde{C}d\log(T) \equiv C(T).$$

Besides, since $\alpha_t, \alpha^*, \theta_t, \theta^*, X_t, \vec{p_t}$ are bounded by constants depending only on $W, L, K$, there is a constant $C_0$ depending only on $W, L, K$ such that $\sum_{t=1}^{T}\sum_{j=1}^{k_t}((\theta_t - \theta^*)^T x_{t,j} - (\alpha_t - \alpha^*)^T x_{t,j}p_{t,j})^2 \leq C_0 KT$.

Let $\mathbb{A}_T$ denote the event that $\sum_{t=1}^{T}\sum_{j=1}^{k_t}((\theta_t - \theta^*)^T x_{t,j} - (\alpha_t - \alpha^*)^T x_{t,j}p_{t,j})^2 \leq C(T)$.

We have that:

$$\mathbb{E}\left[\sum_{t=1}^{T}\sum_{j=1}^{k_t}((\theta_t - \theta^*)^T x_{t,j} - (\alpha_t - \alpha^*)^T x_{t,j}p_{t,j})^2\right]$$

$$= \mathbb{E}\left[\mathbf{1}_{\mathbb{A}_T}\sum_{t=1}^{T}\sum_{j=1}^{k_t}((\theta_t - \theta^*)^T x_{t,j} - (\alpha_t - \alpha^*)^T x_{t,j}p_{t,j})^2\right]$$

$$+ \mathbb{E}\left[\mathbf{1}_{\overline{\mathbb{A}_T}}\sum_{t=1}^{T}\sum_{j=1}^{k_t}((\theta_t - \theta^*)^T x_{t,j} - (\alpha_t - \alpha^*)^T x_{t,j}p_{t,j})^2\right]$$

$$\leq C(T) + C_0 KT \times \frac{\log_2(T)}{T^2} \equiv \tilde{C}(T)$$

This implies

$$\mathbb{E}\left[\sum_{t=1}^{T}\sum_{j=1}^{k_t}((\theta_t - \theta^*)^T x_{t,j} - (\alpha_t - \alpha^*)^T x_{t,j}(g(X_t\alpha_t, X_t\theta_t)_j + \Delta p_{t,j}))^2\right] \leq \tilde{C}(T).$$

Given that for all $t, j$, $\Delta p_{t,j}$ has zero mean, developing the above sum implies the following inequalities:

$$\mathbb{E} \sum_{t=1}^{T} \sum_{j=1}^{k_t} ((x_{t,j}^\top(\theta_t - \theta^*)) - g(X_t \alpha_t, X_t \theta_t)_j (x_{t,j}^\top(\alpha_t - \alpha^*)))^2 \leq \tilde{C}(T) \tag{10}$$

and

$$\mathbb{E} \sum_{t=1}^{T} \sum_{j=1}^{k_t} ((\alpha_t - \alpha^*)^T x_{t,j})^2 \Delta p_{t,j}^2 \leq \tilde{C}(T). \tag{11}$$

We now use the two above inequalities to provide bounds on $\mathbb{E} \sum_{t=1}^{T} \sum_{j=1}^{k_t} ((\alpha_t - \alpha^*)^T x_{t,j})^2$ and on $\mathbb{E} \sum_{t=1}^{T} \sum_{j=1}^{k_t} ((\theta_t - \theta^*)^T x_{t,j})^2$.

First, since for all $t \leq T$, $\Delta p_{t,j}^2 \geq \frac{1}{W^2 \sqrt{T}}$, it follows that

$$\mathbb{E} \sum_{t=1}^{T} \sum_{j=1}^{k_t} ((\alpha_t - \alpha^*)^T x_{t,j})^2 \leq W^2 \tilde{C}(T) \sqrt{T}. \tag{12}$$

Then, using the Cauchy-Schwartz inequality and noting that $g(X_t \alpha_t, X_t \theta_t)_j \leq p_{\max}$ by Lemma B.2, we have

$$\mathbb{E} \left( \sum_{t=1}^{T} \sum_{j=1}^{k_t} 2 g(X_t \alpha_t, X_t \theta_t)_j x_{t,j}^\top (\theta_t - \theta^*) x_{t,j}^\top (\alpha_t - \alpha^*) \right)$$

$$\leq 2 p_{\max} \mathbb{E} \left( \sum_{t=1}^{T} \sum_{j=1}^{k_t} x_{t,j}^\top (\theta_t - \theta^*) x_{t,j}^\top (\alpha_t - \alpha^*) \right)$$

$$\leq 2 p_{max} \sqrt{\mathbb{E} \left[ \sum_{t=1}^{T} \sum_{j=1}^{k_t} x_{t,j}^\top (\alpha_t - \alpha^*)^2 \right]} \sqrt{\mathbb{E} \left[ \sum_{t=1}^{T} \sum_{j=1}^{k_t} x_{t,j}^\top (\theta_t - \theta^*)^2 \right]}$$

$$\leq 2 p_{max} T^{1/4} \sqrt{W^2 \tilde{C}(T)} \sqrt{\mathbb{E} \left[ \sum_{t=1}^{T} \sum_{j=1}^{k_t} x_{t,j}^\top (\theta_t - \theta^*)^2 \right]}.$$

Hence, first noting that (10) implies that

$$\mathbb{E} \left[ \sum_{t=1}^{T} \sum_{j=1}^{k_t} x_{t,j}^\top (\theta_t - \theta^*)^2 \right] - \mathbb{E} \left( \sum_{t=1}^{T} \sum_{j=1}^{k_t} 2 g(X_t \alpha_t, X_t \theta_t)_j x_{t,j}^\top (\theta_t - \theta^*) x_{t,j}^\top (\alpha_t - \alpha^*) \right) \leq \tilde{C}(T),$$

we obtain

$$\mathbb{E} \left[ \sum_{t=1}^{T} \sum_{j=1}^{k_t} x_{t,j}^\top (\theta_t - \theta^*)^2 \right] - 2 p_{max} T^{1/4} \sqrt{W^2 \tilde{C}(T)} \sqrt{\mathbb{E} \left[ \sum_{t=1}^{T} \sum_{j=1}^{k_t} x_{t,j}^\top (\theta_t - \theta^*)^2 \right]} \leq \tilde{C}(T). \tag{13}$$

Let $A := \mathbb{E} \left[ \sum_{t=1}^{T} \sum_{j=1}^{k_t} x_{t,j}^\top (\theta_t - \theta^*)^2 \right]$. Consider the two following cases:

- $2 p_{max} T^{1/4} \sqrt{W^2 \tilde{C}(T)} \sqrt{A} \leq \frac{A}{2}$. Then, from (13), we obtain that $A \leq 2 \tilde{C}(T)$.

- $2 p_{max} T^{1/4} \sqrt{W^2 \tilde{C}(T)} \sqrt{A} > \frac{A}{2}$. Then $A \leq 16 p_{max}^2 \sqrt{T} W^2 \tilde{C}(T)$.

So $A \leq 16 \max(2, p_{max})^2 \sqrt{T} W^2 \tilde{C}(T)$.

Coming back to the upper bound on the regret given by equation (9), we obtain that the total regret is bounded as follows:

$$R^\pi(T) \leq C_1 C_2 \left[ \sqrt{T} W^2 \tilde{C}(T)(1 + 16 \max(2, p_{max})^2) + W^2 \tilde{C}(T) \sqrt{T} \right] + \frac{2 C_1 K \sqrt{T}}{W^2}.$$

Using the definition of $\tilde{C}(T) = \tilde{C} d \log(T)$, we conclude that for some constant $C > 0$ depending on W,K,L,

$$R^\pi(T) \leq C d \log(T) \sqrt{T}.$$

$\square$

## B.2  Proof of the main lemmas

Throughout this section, we will use the following closed form expressions of the gradient and hessian of the loss $\ell_{t,1}$.

Remember that $y_{t,j}$ is the binary variable which indicates whether the customer has purchased product $j$ at time t. The loss at time $t$ can be expressed as:

$$\ell_{t,1}(\gamma) = \sum_{j=0}^{k_t} -y_{t,j} \log\left(q_{t,j}(\gamma, \vec{p_t})\right) = \log\left(1 + \sum_{j=1}^{k_t} e^{\tilde{x}_{t,j}^\top \gamma}\right) - \sum_{j=1}^{k_t} y_{t,i} \tilde{x}_{t,i}^\top \gamma$$

We can then write the gradient and hessian of $\ell_{t,1}$ as:

$$\nabla \ell_{t,1}(\gamma) = \sum_{j=1}^{k_t} (q_{t,j}(\gamma, \vec{p_t}) - y_{t,j}) \tilde{x}_{t,j} \tag{14}$$

and

$$\nabla^2 \ell_{t,1}(\gamma) = \frac{\sum_{j=1}^{k_t} e^{\tilde{x}_{t,j}^\top \gamma} \tilde{x}_{t,j} \tilde{x}_{t,j}^T (1 + \sum_{j=1}^{k_t} e^{\tilde{x}_{t,j}^\top \gamma}) - (\sum_{j=1}^{k_t} e^{\tilde{x}_{t,j}^\top \gamma} \tilde{x}_{t,j})(\sum_{j=1}^{k_t} e^{\tilde{x}_{t,j}^\top \gamma} \tilde{x}_{t,j}^T)}{(1 + \sum_{j=1}^{k_t} e^{\tilde{x}_{t,j}^\top \gamma})^2}$$

$$= \sum_{j=1}^{k_t} q_{t,j}(\gamma, \vec{p_t}) \tilde{x}_{t,j} \tilde{x}_{t,j}^T - \sum_{j=1}^{k_t} \sum_{i=1}^{k_t} q_{t,j}(\gamma, \vec{p_t}) q_{t,i}(\gamma, \vec{p_t}) \tilde{x}_{t,j} \tilde{x}_{t,i}^T. \tag{15}$$

**Lemma B.5.** *All following properties hold for all $t \geq 0$ and $\gamma \in \mathbb{R}^d$:*

1. $\nabla^2 \ell_{t,1}(\gamma) \succeq \kappa_1 \sum_{j=1}^{k_t} \tilde{x}_{t,j} \tilde{x}_{t,j}^T$,

2. $\nabla^2 \ell_{t,1}(\gamma) \preceq 2 \sum_{j=1}^{k_t} \tilde{x}_{t,j} \tilde{x}_{t,j}^T$,

3. $\nabla \ell_{t,1}(\gamma) \nabla \ell_{t,1}(\gamma)^T \preceq 2 \sum_{j=1}^{k_t} \tilde{x}_{t,j} \tilde{x}_{t,j}^T$,

4. $\nabla \ell_{t,1}(\gamma)^T \nabla \ell_{t,1}(\gamma) \leq 2 \sum_{j=1}^{k_t} \tilde{x}_{t,j}^T \tilde{x}_{t,j}$.

*Proof.* First note that for all $i, j \in [k_t]$:

$$(\tilde{x}_{t,j} - \tilde{x}_{t,i})^T (\tilde{x}_{t,j} - \tilde{x}_{t,i}) = \tilde{x}_{t,j} \tilde{x}_{t,j}^T + \tilde{x}_{t,i} \tilde{x}_{t,i}^T - \tilde{x}_{t,j} \tilde{x}_{t,i}^T - \tilde{x}_{t,i} \tilde{x}_{t,j}^T \succeq 0$$

$$(\tilde{x}_{t,j} + \tilde{x}_{t,i})^T (\tilde{x}_{t,j} + \tilde{x}_{t,i}) = \tilde{x}_{t,j} \tilde{x}_{t,j}^T + \tilde{x}_{t,i} \tilde{x}_{t,i}^T + \tilde{x}_{t,j} \tilde{x}_{t,i}^T + \tilde{x}_{t,i} \tilde{x}_{t,j}^T \succeq 0$$

Hence for all $\alpha \in \mathbb{R}$,

$$|\alpha|(\tilde{x}_{t,j} \tilde{x}_{t,j}^T + \tilde{x}_{t,i} \tilde{x}_{t,i}^T) \succeq \alpha(\tilde{x}_{t,j} \tilde{x}_{t,i}^T + \tilde{x}_{t,i} \tilde{x}_{t,j}^T). \tag{16}$$

Similarly, for all $\alpha \in \mathbb{R}$,

$$|\alpha|(\tilde{x}_{t,j}^T \tilde{x}_{t,j} + \tilde{x}_{t,i}^T \tilde{x}_{t,i}) \geq \alpha(\tilde{x}_{t,j}^T \tilde{x}_{t,i} + \tilde{x}_{t,i}^T \tilde{x}_{t,j}). \tag{17}$$

It follows that:

1.
$$\nabla^2 \ell_{t,1}(\gamma) = \sum_{j=1}^{k_t} q_{t,j}(\gamma,\vec{p_t})\tilde{x}_{t,j}\tilde{x}_{t,j}^T - \sum_{j=1}^{k_t}\sum_{i=1}^{k_t} q_{t,j}(\gamma,\vec{p_t})q_{t,i}(\gamma,\vec{p_t})\tilde{x}_{t,j}\tilde{x}_{t,i}^T$$

$$= \sum_{j=1}^{k_t} q_{t,j}(\gamma,\vec{p_t})\tilde{x}_{t,j}\tilde{x}_{t,j}^T - \frac{1}{2}\sum_{j=1}^{k_t}\sum_{i=1}^{k_t} q_{t,j}(\gamma,\vec{p_t})q_{t,i}(\gamma,\vec{p_t})(\tilde{x}_{t,j}\tilde{x}_{t,i}^T + \tilde{x}_{t,i}\tilde{x}_{t,j}^T)$$

$$\succeq \sum_{j=1}^{k_t} q_{t,j}(\gamma,\vec{p_t})\tilde{x}_{t,j}\tilde{x}_{t,j}^T - \frac{1}{2}\sum_{j=1}^{k_t}\sum_{i=1}^{k_t} q_{t,j}(\gamma,\vec{p_t})q_{t,i}(\gamma,\vec{p_t})(\tilde{x}_{t,i}\tilde{x}_{t,i}^T + \tilde{x}_{t,j}\tilde{x}_{t,j}^T)$$

$$= \sum_{j=1}^{k_t} q_{t,j}(\gamma,\vec{p_t})\left(1 - \sum_{i=1}^{k_t} q_{t,j}(\gamma,\vec{p_t})\right)\tilde{x}_{t,j}\tilde{x}_{t,j}^T$$

$$= \sum_{j=1}^{k_t} q_{t,j}(\gamma,\vec{p_t})q_{t,0}(\gamma,\vec{p_t})\tilde{x}_{t,j}\tilde{x}_{t,j}^T$$

$$\succeq \kappa_1 \sum_{j=1}^{k_t} \tilde{x}_{t,j}\tilde{x}_{t,j}^T.$$

2.
$$\nabla^2 \ell_{t,1}(\gamma_t) = \sum_{j=1}^{k_t} q_{t,j}(\gamma,\vec{p_t})\tilde{x}_{t,j}\tilde{x}_{t,j}^T - \sum_{j=1}^{k_t}\sum_{i=1}^{k_t} q_{t,j}(\gamma,\vec{p_t})q_{t,i}(\gamma,\vec{p_t})\tilde{x}_{t,j}\tilde{x}_{t,i}^T$$

$$\preceq \sum_{j=1}^{k_t} q_{t,j}(\gamma,\vec{p_t})\tilde{x}_{t,j}\tilde{x}_{t,j}^T + \frac{1}{2}\sum_{j=1}^{k_t}\sum_{i=1}^{k_t} q_{t,j}(\gamma,\vec{p_t})q_{t,i}(\gamma,\vec{p_t})(\tilde{x}_{t,i}\tilde{x}_{t,i}^T + \tilde{x}_{t,j}\tilde{x}_{t,j}^T)$$

$$= \sum_{j=1}^{k_t} q_{t,j}(\gamma,\vec{p_t})\left(1 + \sum_{i=1}^{k_t} q_{t,j}(\gamma,\vec{p_t})\right)\tilde{x}_{t,j}\tilde{x}_{t,j}^T$$

$$\preceq 2\sum_{j=1}^{k_t} \tilde{x}_{t,j}\tilde{x}_{t,j}^T.$$

3.
$$\nabla \ell_{t,1}(\gamma_t)\nabla \ell_{t,1}(\gamma_t)^T = \sum_{j=1}^{k_t}\sum_{i=1}^{k_t}(q_{t,j}(\gamma,\vec{p_t}) - y_{t,j})(q_{t,i}(\gamma,\vec{p_t}) - y_{t,i})\tilde{x}_{t,j}\tilde{x}_{t,i}^T$$

$$= \frac{1}{2}\sum_{j=1}^{k_t}\sum_{i=1}^{k_t}(q_{t,j}(\gamma,\vec{p_t}) - y_{t,j})(q_{t,i}(\gamma,\vec{p_t}) - y_{t,i})(\tilde{x}_{t,j}\tilde{x}_{t,i}^T + \tilde{x}_{t,i}\tilde{x}_{t,j}^T)$$

$$\preceq \frac{1}{2}\sum_{j=1}^{k_t}\sum_{i=1}^{k_t}|q_{t,j}(\gamma,\vec{p_t}) - y_{t,j}||q_{t,i}(\gamma,\vec{p_t}) - y_{t,i}|(\tilde{x}_{t,j}\tilde{x}_{t,j}^T + \tilde{x}_{t,i}\tilde{x}_{t,i}^T)$$

by (16)

$$= \sum_{j=1}^{k_t}|q_{t,j}(\gamma,\vec{p_t}) - y_{t,j}|\tilde{x}_{t,j}\tilde{x}_{t,j}^T \sum_{i=1}^{k_t}|q_{t,i}(\gamma,\vec{p_t}) - y_{t,i}|$$

$$\preceq 2\sum_{j=1}^{k_t} \tilde{x}_{t,j}\tilde{x}_{t,j}^T,$$

where the last inequality uses that for all $i \in [k_t]$, $|q_{t,i}(\gamma, \vec{p_t}) - y_{t,i}| \le 1$ and that

$$\sum_{i=1}^{k_t} |q_{t,i}(\gamma, \vec{p_t}) - y_{t,i}| = \begin{cases} \sum_{i \in [k_t] \setminus \{i_t\}} q_{t,i}(\gamma, \vec{p_t}) + (1 - q_{t,i_t}(\gamma, \vec{p_t})) & \text{if } i_t \in [k_t] \\ \sum_{i \in [k_t] \setminus \{i_t\}} q_{t,i}(\gamma, \vec{p_t}) & \text{if } i_t = 0 \end{cases} \le 2.$$

4. Similarly,

$$\nabla \ell_{t,1}(\gamma_t)^T \nabla \ell_{t,1}(\gamma_t)^T = \sum_{j=1}^{k_t} \sum_{i=1}^{k_t} (q_{t,j}(\gamma, \vec{p_t}) - y_{t,j})(q_{t,i}(\gamma, \vec{p_t}) - y_{t,i}) \tilde{x}_{t,j}^T \tilde{x}_{t,i}$$

$$\le \frac{1}{2} \sum_{j=1}^{k_t} \sum_{i=1}^{k_t} |q_{t,j}(\gamma, \vec{p_t}) - y_{t,j}||q_{t,i}(\gamma, \vec{p_t}) - y_{t,i}|(\tilde{x}_{t,j}^T \tilde{x}_{t,j} + \tilde{x}_{t,i}^T \tilde{x}_{t,i})$$

by (17)

$$\le 2 \sum_{j=1}^{k_t} \tilde{x}_{t,j} \tilde{x}_{t,j}^T.$$

$\square$

## B.3 Proof of Lemma 2.8

In order to prove Lemma 2.8, we will combine the lower and upper bounds on $\sum_{t=1}^{T}[\ell_{t,1}(\gamma_t) - \ell_{t,1}(\gamma^*)]$ provided in the two following lemmas. The proofs are deferred to Appendices B.4 and B.5, respectively.

**Lemma B.6** (Upper bound $\sum_{t=1}^{T} \ell_{t,1}(\gamma_t) - \sum_{t=1}^{T} \ell_{t,1}(\gamma^*)$)**.**

$$\sum_{t=1}^{T}[\ell_{t,1}(\gamma_t) - \ell_{t,1}(\gamma^*)] \le \frac{2d}{\kappa_1} \log\left(\lambda_T + \frac{2TK}{d}\right) + 2\mu W^2 - \frac{\mu \kappa_1}{2} \sum_{t=1}^{T} \sum_{j=1}^{k_t} ((\gamma_t - \gamma^*)^T \tilde{x}_{t,j})^2$$

**Lemma B.7** (Lower bound $\sum_{t=1}^{T} \ell_{t,1}(\gamma_t) - \sum_{t=1}^{T} \ell_{t,1}(\gamma^*)$)**.**

$$\mathbb{P}\left(\sum_{t=1}^{T}[\ell_{t,1}(\gamma_t) - \ell_{t,1}(\gamma^*)] \le -2\sqrt{2\log T \sum_{t=1}^{T} \sum_{j=1}^{k_t}((\gamma_t - \gamma^*)^T \tilde{x}_{t,j})^2}\right.$$

$$\left. - KW(1 + p_{max})\left(\frac{4\log(T)}{3} + 1\right)\right) \le \frac{\lceil 2\log_2(T) + 1 \rceil}{T^2}. \quad (18)$$

We now prove the following expanded version of Lemma 2.8.

*Lemma 2.8* 1. With probability at least $1 - \frac{\log_2(T)}{T^2}$,

$$\sum_{t=1}^{T} \sum_{j=1}^{k_t} (x_{t,j}^\top(\theta_t - \theta^*) - x_{t,j}^\top(\alpha_t - \alpha^*)p_{t,j})^2 \le C(T) := \max\left\{\frac{128\log(T)}{\kappa_1^2 \mu^2},\right.$$

$$\left. \frac{4}{\mu\kappa_1}\left(\frac{2d}{\kappa_1}\log\left(\lambda_T + \frac{2TK}{d}\right) + 2\mu W^2 + KW(1 + p_{max})\left(\frac{4\log(T)}{3} + 1\right)\right)\right\}$$

Using that $\lambda_t = d \log(t)$ for all $t \ge 2$, we get that for a constant $\tilde{C}$ depending only on $W, L, K$:

$$\sum_{t=1}^{T} \sum_{j=1}^{k_t} (x_{t,j}^\top(\theta_t - \theta^*) - x_{t,j}^\top(\alpha_t - \alpha^*)p_{t,j})^2 \le \tilde{C}d \log(T).$$

*Proof.* To ease the presentation, let

$$B(T) := \frac{2d}{\kappa_1} \log\left(\lambda_T + \frac{2TK}{d}\right) + 2\mu W^2 + KW(1 + p_{max})\left(\frac{4\log(T)}{3} + 1\right)$$

We obtain by combining Lemmas B.6 and B.7 that with probability at least $1 - \frac{\lceil\log(T)\rceil}{T^2}$:

$$\frac{\mu\kappa_1}{2}\sum_{t=1}^{T}\sum_{j=1}^{k_t}((\gamma_t - \gamma^*)^T\tilde{x}_{t,j})^2 - 2\sqrt{2\log T}\left\{\sum_{t=1}^{T}\sum_{j=1}^{k_t}((\gamma_t - \gamma^*)^T\tilde{x}_{t,j})^2\right\}^{1/2} \leq B(T) \quad (19)$$

Now let $A = \sum_{t=1}^{T}\sum_{j=1}^{k_t}((\gamma_t - \gamma^*)^T\tilde{x}_{t,j})^2$ and consider the two following cases:

- Suppose $\frac{1}{2}\frac{\mu\kappa_1 A}{2} \geq 2\sqrt{2\log(T)}\sqrt{A}$. Then from equation (19), we obtain that: $A \leq \frac{4B(T)}{\mu\kappa_1}$.

- Else, $\frac{1}{2}\frac{\mu\kappa_1 A}{2} < 2\sqrt{2\log(T)}\sqrt{A}$ hence by reorganizing the terms, we get $A \leq \frac{128\log(T)}{\kappa_1^2\mu^2}$.

The proof is complete. $\qquad\qquad\qquad\qquad\qquad\qquad\qquad\qquad\qquad\qquad\qquad\qquad\qquad\square$

## B.4  Proof of Lemma B.6

*Proof.* Consider $\hat{\gamma}_{t+1} = \gamma_t - \frac{1}{\mu}H_{T+1}^{-1}\nabla\ell_{t,1}(\gamma_t)$, so that $\gamma_{t+1}$ is the projection of $\hat{\gamma}_{t+1}$ in the norm induced by $H_t$.

By doing a Taylor expansion of $\ell_{t,1}$, we obtain:

$$\ell_{t,1}(\gamma_t) - \ell_{t,1}(\gamma^*) = \nabla\ell_{t,1}(\gamma_t)^T(\gamma_t - \gamma^*)$$
$$- (\gamma_t - \gamma^*)^T\int_0^1\int_0^1 z\nabla^2\ell_{t,1}(\gamma_t + zw(\gamma^* - \gamma_t))dzdw(\gamma_t - \gamma^*)$$

Using Proposition A.4 and the definition of $\mu = \frac{1}{2(1+(1+p_{max})2\sqrt{6KW})}$, this leads to:

$$\ell_{t,1}(\gamma_t) - \ell_{t,1}(\gamma^*) \leq \nabla\ell_{t,1}(\gamma_t)^T(\gamma_t - \gamma^*) - \mu(\gamma_t - \gamma^*)^T\nabla^2\ell_{t,1}(\gamma_t)(\gamma_t - \gamma^*) \quad (20)$$

Note that in the classical Online Newton Step analysis, a similar equation as the above equation is used, but with the potentially exponentially small exp-concavity parameter $\beta$ instead of $\mu$. The next part of the proof globally follows the ONS analysis in [20]. We include it here for completeness.

By definition of $\hat{\gamma}_{t+1}$, we can write the following two equalities:

$$\hat{\gamma}_{t+1} - \gamma^* = \gamma_t - \gamma^* - \frac{1}{\mu}H_t^{-1}\nabla\ell_{t,1}(\gamma_t)$$

and

$$H_t(\hat{\gamma}_{t+1} - \gamma^*) = H_t(\gamma_t - \gamma^*) - \frac{1}{\mu}\nabla\ell_{t,1}(\gamma_t).$$

Hence, by multiplying the transpose of the first inequality with the second inequality, we obtain

$$(\hat{\gamma}_{t+1} - \gamma^*)^T H_t(\hat{\gamma}_{t+1} - \gamma^*)$$
$$= (\gamma_t - \gamma^*)^T H_t(\gamma_t - \gamma^*) - \frac{2}{\mu}\nabla\ell_{t,1}^T(\gamma_t)(\gamma_t - \gamma^*) + \frac{1}{\mu^2}\nabla\ell_{t,1}(\gamma_t)^T H_t^{-1}\nabla\ell_{t,1}(\gamma_t)$$

Since $\gamma_{t+1}$ is the projection of $\hat{\gamma}_{t+1}$ on $B_{t+1}$ relatively to the norm induced by $H_t$, we have the following inequality:

$$(\hat{\gamma}_{t+1} - \gamma^*)^T H_t(\hat{\gamma}_{t+1} - \gamma^*)^T \geq (\gamma_{t+1} - \gamma^*)^T H_t(\gamma_{t+1} - \gamma^*) \quad (21)$$

Combining equations (**??**) and (21) gives the following bound on $\nabla\ell_{t,1}(\gamma_t)^T(\gamma_t - \gamma^*)$:

$$\nabla\ell_{t,1}(\gamma_t)^T(\gamma_t - \gamma^*)$$
$$\leq \frac{1}{2\mu}\nabla\ell_{t,1}(\gamma_t)^T H_t^{-1}\nabla\ell_{t,1}(\gamma_t) + \frac{\mu}{2}(\gamma_t - \gamma^*)H_t(\gamma_t - \gamma^*) - \frac{\mu}{2}(\gamma_{t+1} - \gamma^*)^T H_t(\gamma_{t+1} - \gamma^*)$$

Hence, summing from $t = 1$ to $T$:

$$\sum_{t=1}^{T} \nabla \ell_{t,1}(\gamma_t)^T (\gamma_t - \gamma^*) \leq \frac{1}{2\mu} \sum_{t=1}^{T} \nabla \ell_{t,1}(\gamma_t)^T H_t^{-1} \nabla \ell_{t,1}(\gamma_t)$$
$$+ \frac{\mu}{2}(\gamma_1 - \gamma^*) H_1 (\gamma_1 - \gamma^*) - \frac{\mu}{2}(\gamma_{T+1} - \gamma^*) H_{T+1}(\gamma_{T+1} - \gamma^*)$$
$$+ \frac{\mu}{2} \sum_{t=2}^{T} (\gamma_t - \gamma^*)^T (H_t - H_{t-1})(\gamma_t - \gamma^*)$$
$$\leq \frac{1}{2\mu} \sum_{t=1}^{T} \nabla \ell_{t,1}(\gamma_t)^T H_t^{-1} \nabla \ell_{t,1}(\gamma_t)$$
$$+ \frac{\mu}{2}(\gamma_1 - \gamma^*)^T (H_1 - \nabla^2 l_1(\gamma_1))(\gamma_1 - \gamma^*)$$
$$+ \frac{\mu}{2} \sum_{t=1}^{T} (\gamma_t - \gamma^*)^T (H_t - H_{t-1})(\gamma_t - \gamma^*)$$

In the last part of the proof, we bring out more specifically the sum $\sum_{t=1}^{T} \sum_{j=1}^{k_t} ((\gamma_t - \gamma^*)^T \tilde{x}_{t,j})^2$. Since $H_t - H_{t-1} = \nabla^2 \ell_{t,1}(\gamma_t)$, we obtain, by combining the above inequality with (20) and by using the lower bound on $\nabla \ell_{t,1}^2$ provided in Lemma B.5:

$$\sum_{t=1}^{T} \ell_{t,1}(\gamma_t) - \ell_{t,1}(\gamma^*)$$
$$\leq \sum_{t=1}^{T} \nabla \ell_{t,1}(\gamma_t)^T (\gamma_t - \gamma^*) - \mu \sum_{t=1}^{T} (\gamma_t - \gamma^*)^T \nabla^2 \ell_{t,1}(\gamma_t)(\gamma_t - \gamma^*)$$
$$\leq \frac{1}{2\mu} \sum_{t=1}^{T} \nabla \ell_{t,1}(\gamma_t)^T H_t^{-1} \nabla \ell_{t,1}(\gamma_t) + \frac{\mu}{2}(\gamma_1 - \gamma^*)^T (H_1 - \nabla^2 l_1(\gamma_1))(\gamma_1 - \gamma^*)$$
$$- \frac{\mu}{2} \sum_{t=1}^{T} (\gamma_t - \gamma^*)^T \nabla^2 \ell_{t,1}(\gamma_t)(\gamma_t - \gamma^*)$$
$$\leq \frac{1}{2\mu} \sum_{t=1}^{T} \nabla \ell_{t,1}(\gamma_t)^T H_t^{-1} \nabla \ell_{t,1}(\gamma_t) + \frac{\mu}{2}(\gamma_1 - \gamma^*)^T (H_1 - \nabla^2 l_1(\gamma_1))(\gamma_1 - \gamma^*)$$
$$- \frac{\mu \kappa_1}{2} \sum_{t=1}^{T} \sum_{j=1}^{k_t} ((\gamma_t - \gamma^*)^T \tilde{x}_{t,j})^2$$

Applying Lemma B.8 (stated below) and noting that $\frac{\mu}{2}(\gamma_1 - \gamma^*)^T (H_1 - \nabla^2 l_1(\gamma_1))(\gamma_1 - \gamma^*) = \frac{\mu}{2}(\gamma_1 - \gamma^*)^T (\lambda_1 I_d)(\gamma_1 - \gamma^*) = \frac{\mu \lambda_1 \|\gamma_1 - \gamma^*\|^2}{2} \leq \frac{\mu \lambda_1 (2W)^2}{2} = 2\mu W^2$ concludes the proof of the lemma.

$\square$

**Lemma B.8.**

$$\sum_{t=1}^{T} \nabla \ell_{t,1}(\gamma_t)^T H_t^{-1} \nabla \ell_{t,1}(\gamma_t) \leq \frac{2d}{\kappa_1} \log \left( \lambda_T + \frac{2TK}{d} \right)$$

*Proof.* By definition of $H_t$:

$$H_{t+1} = H_t + \nabla^2 \ell_{t+1,1}(\gamma_{t+1}) + (\lambda_{t+1} - \lambda_t) I_d$$

$$\succeq H_t + \kappa_1 \sum_{j=1}^{k_{t+1}} \tilde{x}_{t+1,j} \tilde{x}_{t+1,j}^T \qquad \text{(by Lemma B.5 and using } \lambda_{t+1} \geq \lambda_t)$$

$$\succeq H_t + \frac{\kappa_1}{2} \nabla \ell_{t+1,1}(\gamma_{t+1}) \nabla \ell_{t+1,1}(\gamma_{t+1})^T. \qquad \text{(by Lemma B.5)}$$

The proof now globally uses similar techniques as in the proof of Lemma 11.11 in [10]. From above, we obtain

$$\det(H_{t+1}) \cdot \det \left( I_d - \frac{\kappa_1}{2} H_{t+1}^{-1/2} \nabla \ell_{t+1,1}(\gamma_{t+1}) \nabla \ell_{t+1,1}(\gamma_{t+1})^T H_{t+1}^{-1/2} \right) \geq \det(H_t). \qquad (22)$$

Using that for all $z \in \mathbb{R}^d$, $\det(1 - zz^T) = 1 - z^T z$ (see [10], Lemma 11.11), and that $H_{t+1}$ is symmetric, we get

$$\det \left( I_d - \frac{\kappa_1}{2} H_{t+1}^{-1/2} \nabla \ell_{t+1,1}(\gamma_{t+1}) \nabla \ell_{t+1,1}(\gamma_{t+1})^T H_{t+1}^{-1/2} \right)$$

$$= \det \left( I_d - \left( \sqrt{\frac{\kappa_1}{2}} H_{t+1}^{-1/2} \nabla \ell_{t+1,1}(\gamma_{t+1}) \right) \left( \sqrt{\frac{\kappa_1}{2}} H_{t+1}^{-1/2} \nabla \ell_{t+1,1}(\gamma_{t+1}) \right)^T \right)$$

$$= 1 - \frac{\kappa_1}{2} \nabla \ell_{t+1,1}(\gamma_{t+1})^T H_{t+1}^{-1} \nabla \ell_{t+1,1}(\gamma_{t+1}).$$

Hence, combining this with (22) and reorganizing the terms, we obtain

$$\frac{\kappa_1}{2} \nabla \ell_{t+1,1}(\gamma_{t+1})^T H_{t+1}^{-1} \nabla \ell_{t+1,1}(\gamma_{t+1}) \leq 1 - \frac{\det(H_t)}{\det(H_{t+1})}.$$

Summing from $t = 0$ to $T - 1$ and translating the indices gives:

$$\frac{\kappa_1}{2} \sum_{t=1}^{T} \nabla \ell_{t,1}(\gamma_t)^T H_t^{-1} \nabla \ell_{t,1}(\gamma_t) \leq \sum_{t=1}^{T} \left( 1 - \frac{\det(H_{t-1})}{\det(H_t)} \right)$$

$$\leq \sum_{t=1}^{T} \log \left( \frac{\det(H_t)}{\det(H_{t-1})} \right) \qquad (1 - x \leq -\log(x) \text{ for all } x > 0)$$

$$= \log \left( \frac{\det(H_T)}{\det(\lambda_1 I_d)} \right). \qquad (23)$$

We now bound $\log(\det(H_T))$. First note that for all $t \in [T]$, we have by Lemma B.5 that $\nabla^2 \ell_{t,1}(\gamma_t) \preceq 2 \sum_{j=1}^{k_t} \tilde{x}_{t,j} \tilde{x}_{t,j}^T$. Since $||x_{t,j}||_2 \leq 1$ for all $t \in [T], j \in [k_t]$, it follows that:

$$\text{trace}(H_T) = \text{trace} \left( \lambda_T I_d + \sum_{t=1}^{T} \nabla^2 \ell_{t,1}(\gamma_t) \right)$$

$$\leq \text{trace}(\lambda_T I_d) + 2 \sum_{t=1}^{T} \sum_{j=1}^{k_t} \text{trace}\left( \tilde{x}_{t,j} \tilde{x}_{t,j}^T \right)$$

$$= \text{trace}(\lambda_T I_d) + 2 \sum_{t=1}^{T} \sum_{j=1}^{k_t} \text{trace}\left( \tilde{x}_{t,j}^T \tilde{x}_{t,j} \right)$$

$$\leq d\lambda_T + 2TK.$$

Hence, using the determinant-trace inequality $\det(H_T) \leq \left( \frac{\text{trace}(H_T)}{d} \right)^d$ (see [10]) and $\lambda_1 = 1$, we obtain that:

$$\log \left( \frac{\det(H_T)}{\det(\lambda_1 I_d)} \right) = \log \left( \frac{\det(H_T)}{\lambda_1^d} \right) \leq d \log \left( \lambda_T + \frac{2TK}{d} \right) \qquad (24)$$

Putting (23) and (24) together concludes the proof of the lemma. $\qquad \square$

## B.5   Proof of Lemma B.7

The proof relies on a Bernstein type inequality for martingales difference sequences from [19]:

**Proposition B.9.** *([19], Th 1.6) Let $X_1, ..., X_n$ be a bounded martingale difference sequence with respect to the filtration $(\mathcal{F}_i)_{1 \leq i \leq n}$ such that $|X_i| \leq M$ for all i.*

*Let $\Sigma_n^2$ denote the sum of the conditional variances.*

$$\Sigma_n^2 = \sum_{i=1}^{n} \mathbb{E}(X_i^2 | \mathcal{F}_{i-1})$$

*Then,*

$$\mathbb{P}(\sum_{i=1}^{n} X_i \geq \epsilon, \Sigma_n^2 \leq k) \leq exp\left(\frac{-\epsilon^2}{2(k + \frac{\epsilon M}{3})}\right).$$

We are now ready to prove Lemma B.7.

*Proof.* By convexity of $\ell_{t,1}$:

$$\ell_{t,1}(\gamma^*) - \ell_{t,1}(\gamma_t) \leq \nabla \ell_{t,1}(\gamma^*)^T(\gamma^* - \gamma_t),$$

Hence, letting $D_t = \nabla \ell_{t,1}(\gamma^*)^T(\gamma^* - \gamma_t)$, we have

$$\mathbb{P}\left(\sum_{t=1}^{T} \ell_{t,1}(\gamma_t) - \sum_{t=1}^{T} \ell_{t,1}(\gamma^*) \leq -2\sqrt{2 \log T \sum_{t=1}^{T} \sum_{j=1}^{k_t}((\gamma_t - \gamma^*)^T \tilde{x}_{t,j})^2} - \frac{4B \log(T)}{3} - B\right)$$

$$\leq \mathbb{P}\left(\sum_{t=1}^{T} D_t \geq 2\sqrt{2 \log T \sum_{t=1}^{T} \sum_{j=1}^{k_t}((\gamma_t - \gamma^*)^T \tilde{x}_{t,j})^2} + \frac{4B \log(T)}{3} + B\right). \quad (25)$$

Let $\mathcal{F}_t$ be the filtration generated by $\{X_1, \Delta \vec{p}_1, z_1, ..., X_t, \Delta \vec{p}_t, z_t, X_{t+1}, \Delta \vec{p}_{t+1}\}$. Since $\gamma_t, \vec{p}_t$ are $\mathcal{F}_{t-1}$ measurable, and using the expression of $\nabla \ell_t$ in (14), and the fact that for all $j \in [k_t]$, $\mathbb{E}[y_{t,j}] = q_{t,j}(\gamma^*, \vec{p}_t)$, we have that:

$$\mathbb{E}(D_t | \mathcal{F}_{t-1}) = \mathbb{E}\left(\sum_{j=1}^{k_t}(q_{t,j}(\gamma^*, \vec{p}_t) - y_{t,j})\tilde{x}_{t,j} \middle| \mathcal{F}_{t-1}\right)^T (\gamma^* - \gamma_t) = 0$$

Therefore, $\{D_t\}_{t=1}^{T}$ is a martingale difference sequence adapted to the filtration $\mathcal{F}_t$.

Moreover, using the Cauchy-Schwartz inequality, we have that $D_t$ is uniformly bounded: $|D_t| = |\sum_{j=1}^{k_t}(q_{t,j}(\gamma^*, \vec{p}_t) - y_{t,j})\tilde{x}_{t,j}^T(\gamma_t - \gamma^*)| \leq \sum_{j=1}^{k_t} |(\gamma_t - \gamma^*)^T \tilde{x}_{t,j}| \leq KW(1 + p_{max})$. We let $B = KW(1 + p_{max})$.

Now, consider $\Sigma_n^2$ the sum of the conditional variances:

$$\Sigma_t^2 = \sum_{i=1}^{t} \mathbb{E}(D_i^2 | \mathcal{F}_{i-1}).$$

By Lemma B.5, we can bound $\Sigma_t^2$ as follows:

$$\Sigma_t^2 = \sum_{s=1}^{t}(\gamma_s - \gamma^*)^T \nabla \ell_{s,1}(\gamma^*)\nabla \ell_{s,1}(\gamma^*)^T(\gamma_s - \gamma^*) \leq \sum_{s=1}^{t} 2 \sum_{j=1}^{k_s}((\gamma_s - \gamma^*)^T \tilde{x}_{s,j})^2 \equiv A_t.$$

Note that we cannot directly apply here the Bernstein inequality from Proposition B.9 with $k = A_t$ since $A_t$ is also a random variable. We address this issue as in [43], making use of a peeling process. First note that by using the Cauchy-Scwhartz inequality, we have $A_t = \sum_{s=1}^{t} 2 \sum_{j=1}^{k_s}((\gamma_s - \gamma^*)^T \tilde{x}_{s,j})^2 \leq 2KTW^2(1 + p_{max})^2 \leq 2B^2T$.

Now, consider two cases:

- $A_T < \frac{B^2}{T}$. Then, using the Cauchy-Schwartz inequality, we get

$$\sum_{t=1}^{T} D_t \le \sqrt{T \sum_{t=1}^{T} D_t^2} \le \sqrt{T(\sum_{t=1}^{T} 2 \sum_{j=1}^{k_t} ((\gamma_t - \gamma^*)^T \tilde{x}_{t,j})^2} < \sqrt{\frac{TB^2}{T}} = B.$$

Thus,

$$\mathbb{P}\left( \sum_{t=1}^{T} D_t \ge \sqrt{2A_T \log(T^2)} + \frac{4B \log(T)}{3} + B \,\middle|\, A_T < \frac{B^2}{T} \right) = 0.$$

- $A_T \ge \frac{B^2}{T}$. Then, since by definition of $B$ and $\Sigma_T^2$, we always have the upper bounds $A_T \le 2B^2 T$ and $\Sigma_T^2 \le A_T$, we have

$$\mathbb{P}\left( \sum_{t=1}^{T} D_t \ge \sqrt{2A_T \log(T^2)} + \frac{2B}{3} \log(T^2) \,\middle|\, A_T \ge \frac{B^2}{T} \right)$$

$$= \mathbb{P}\left( \sum_{t=1}^{T} D_t \ge \sqrt{2A_T \log(T^2)} + \frac{2B}{3} \log(T^2), \Sigma_T^2 \le A_T, \frac{B^2}{T} \le A_T \le 2B^2 T \right)$$

$$\le \sum_{i=1}^{m} \mathbb{P}\left( \sum_{t=1}^{T} D_t \ge \sqrt{\frac{2B^2 2^i log(T^2)}{T}} + \frac{2B}{3} \log(T^2), \Sigma_T^2 \le A_T, \frac{B^2}{T} 2^{i-1} \le A_T \le \frac{B^2}{T} 2^i \right)$$

$$\le \sum_{i=1}^{m} \mathbb{P}\left( \sum_{t=1}^{T} D_t \ge \sqrt{\frac{2B^2 2^i log(T^2)}{T}} + \frac{2B}{3} \log(T^2), \Sigma_T^2 \le \frac{B^2}{T} 2^i \right)$$

$$\le m e^{-log(T^2)}$$

with $m = \lceil \log_2(T^2) \rceil + 1$. The last inequality follows from the Bernstein's inequality for martingales (Proposition B.9) with $k = \frac{B^2}{T} 2^i$ and $\epsilon = \sqrt{2k \log(T^2)} + \frac{2B \log(T^2)}{3}$.

Hence, combining this with (25) we obtain:

$$\mathbb{P}\left( \sum_{t=1}^{T} \ell_{t,1}(\gamma_t) - \sum_{t=1}^{T} \ell_{t,1}(\gamma^*) \le -2\sqrt{2 \log T \sum_{t=1}^{T} \sum_{j=1}^{k_t} ((\gamma_t - \gamma^*)^T \tilde{x}_{t,j})^2} - \frac{4B \log(T)}{3} - B \right)$$

$$\le \mathbb{P}\left( \sum_{t=1}^{T} D_t \ge \sqrt{2A_T \log(T^2)} + \frac{4B \log(T)}{3} + B \right)$$

$$\le \mathbb{P}\left( \sum_{t=1}^{T} D_t \ge \sqrt{2A_T \log(T^2)} + \frac{4B \log(T)}{3} + B \,\middle|\, A_T \ge \frac{B^2}{T} \right) \cdot \mathbb{P}(A_T \ge \frac{B^2}{T}) + 0$$

$$\le \frac{\lceil \log_2(T^2) \rceil + 1}{T^2}. \tag{26}$$

$\square$

## B.6  Proof of Lemmas B.2

*Proof.* Remember that the myopic prices are set as follows: for all $t \ge 1, j \in [k_t]$, $g(X_t \alpha_t, X_t \theta_t)_j = \frac{1}{x_{t,j},^\top \alpha_t} + B_t^0$, where $B_t^0$ is the unique fixed of the following equation:

$$B = \sum_{j=1}^{k_t} \frac{1}{x_{t,j}^\top \alpha_t} e^{-(1+x_{t,j}^\top \alpha_t B)} e^{x_{t,j}^\top \theta_t}. \tag{27}$$

Define the functions $f_1(B) := \sum_{j=1}^{k_t} \frac{1}{x_{t,j}^\top \alpha_t} e^{-(1+x_{t,j}^\top \alpha_t B)} e^{x_{t,j}^\top \theta_t}$ and $f_2(B) := K \frac{1}{L} e^{-(1+LB)+W}$.

By Assumptions 2.2 and 2.3, we have that for for all $B \geq 0$, $f_1(B) \leq K\frac{1}{L}e^{-(1+LB)+W} = f_2(B)$. Now, let $B^u$ be the solution of the following equation:

$$B = f_2(B).$$

Since both $f_1$ and $f_2$ are strictly decreasing, we have that for all $B > B^u$,

$$f_1(B) \leq f_2(B) < f_2(B^u) = B^u < B,$$

thus $B$ is not solution of (27). Hence, we deduce that $B_t^0 \leq B^u$. It follows that the myopic prices are bounded above by $\frac{1}{L} + B^u$.

Next, recall that $B^u$ satisfies

$$B^u = K\frac{1}{L}e^{-(1+LB^u)+W},$$

which, by reorganizing the terms, is equivalent to

$$B^u e^{LB^u} = K\frac{1}{L}e^{-1+W}.$$

Since for $B = K\max(W,1)/L$, we have that $Be^{LB} = \frac{K\max(W,1)}{L}e^{K\max(W,1)} \geq K\frac{1}{L}e^{-1+W}$ and since $B \longmapsto Be^{LB}$ is nondecreasing, we get that $B^u \leq K\max(W,1)/L$. Hence for all $t \geq 1, j \in [k_t]$, $g(X_t\alpha_t, X_t\theta_t)_j \leq \frac{1}{L} + B^u \leq \frac{1+K\max(W,1)}{L}$. Using that $|\Delta p_{t,j}| \leq \frac{1}{W}$, we deduce that $p_{t,j} = g(X_t\alpha_t, X_t\theta_t)_j + \Delta p_{t,j} \leq \frac{1+K\max(W,1)}{L} + \frac{1}{W}$. $\qquad\square$

## C  MNL bandits technical proofs

### C.1  Proof of Theorem 3.4

Before presenting the proof of Theorem 3.4, we need to introduce a few useful lemmas, whose proofs can be found in Appendix C.2. Lemma C.1 is the analogue of the elliptical potential lemma appearing in [1], but uses the local curvature information provided by the terms $\{q_{t,j}(S_t, \theta^*)q_{t,0}(S_t, \theta^*)\}$ to obtain an upper bound that no longer depends on the exponential constant $1/\kappa_2$. In particular, the proof uses the self-concordance-like property of the log-loss.

**Lemma C.1.**

$$\sum_{t=1}^{T}\sum_{j \in S_t} q_{t,j}(S_t, \theta^*)q_{t,0}(S_t, \theta^*)\|x_{t,j}\|_{H_t(\theta^*)^{-1}}^2 \leq 2dK\log\left(\lambda_{T+1} + \frac{2TK}{d}\right)$$

*and*

$$\sum_{t=1}^{T}\sum_{j \in S_t} \|x_{t,j}\|_{H_t(\theta^*)^{-1}}^2 \leq 2d(K + \tfrac{1}{\kappa_2})\log\left(\lambda_{T+1} + \frac{2TK}{d}\right)$$

**Lemma C.2.**

$$\sum_{t=1}^{T}\sum_{j \in S_t} q_{t,j}(S_t, \theta^*)q_{t,0}(S_t, \theta^*) \leq \sum_{t=1}^{T}\kappa_{2,t}^* + R(T)$$

Note that we always have $\sum_{t=1}^{T}\sum_{j \in S_t} q_{t,j}(S_t, \theta^*)q_{t,0}(S_t, \theta^*) \leq T$. In Lemma C.2 we give a tighter upper bound on this sum when the instance is further away from linearity (i.e., when the parameters $\{\kappa_{2,t}^*\}_{t=1}^T$ are small).

**Lemma C.3.** *Define* $Q : \mathbb{R}^K \longrightarrow \mathbb{R}$, *such that for all* $u = \{u_1 \ldots, u_K\} \in \mathbb{R}^n$, $Q(u) = \sum_{i=1}^n \frac{e^{u_i}}{1+\sum_{j=1}^n e^{u_j}}$. *Then for all* $i, j \in [K] \times [K]$,

$$\left|\frac{\partial^2 Q}{\partial i\partial j}\right| \leq 5.$$

**Lemma C.4.** *For all* $\theta_1, \theta_2 \in B(0, W)$

$$\|\theta_1 - \theta_2\|_{H_t(\theta_1)} \leq (1 + \sqrt{6K}W)\|g_t(\theta_1) - g_t(\theta_2)\|_{H_t^{-1}(\theta_1)}.$$

We now give the proof of Theorem 3.4.

*Proof.* Set $\delta = \frac{1}{K^2 T^2}$ and let $A_\delta$ denote the event that $\theta^* \in C_t(\delta)$ for all $t \in [T]$. We know from Proposition 3.3 that $A_\delta$ occurs with probability at least $1 - \delta$. We first assume that $A_\delta$ is satisfied.

Let $\tilde{\theta}^* = (\theta^*, ..., \theta^*) \in \mathbb{R}^{d \times N}$. Since $\theta^* \in C_t(\delta)$, we have by definition of $\tilde{\theta}_{t,j}$ that for all $j \in [k_t]$, $x_{t,j}^\top \tilde{\theta}_{t,j} \geq x_{t,j}^\top \theta^*$, from which we deduce $\tilde{r}_t(S_t^*, \tilde{\theta}_t) \geq \tilde{r}_t(S_t^*, \tilde{\theta}^*)$. Then, by definition of Algorithm 2, the assortment $S_t$ offered at time $t$ satisfies $\tilde{r}_t(S_t, \tilde{\theta}_t) \geq \tilde{r}_t(S_t^*, \tilde{\theta}_t)$ for all $t$. Hence we obtain $\sum_{t=1}^T \sum_{j \in S_t} \tilde{q}_{t,j}(S_t, \tilde{\theta}_t) = \tilde{r}_t(S_t, \tilde{\theta}_t) \geq \tilde{r}_t(S_t^*, \tilde{\theta}^*) = \sum_{t=1}^T \sum_{j \in S_t^*} \tilde{q}_{t,j}(S_t^*, \tilde{\theta}^*) = \sum_{t=1}^T \sum_{j \in S_t^*} q_{t,j}(S_t^*, \theta^*)$, where the last inequality follows by noting that $\tilde{q}_{t,j}(S_t^*, \tilde{\theta}^*) = q_{t,j}(S_t^*, \theta^*)$ for all $t \geq 1, j \in [k_t]$. Hence, by noting that for all $t \geq 1, j \in [k_t]$, we also have that $\tilde{q}_{t,j}(S_t, \tilde{\theta}^*) = q_{t,j}(S_t, \theta^*)$, we can bound the regret as follows:

$$R(T) = \sum_{t=1}^T \left[ \sum_{j \in S_t^*} q_{t,j}(S_t^*, \theta^*) - \sum_{j \in S_t} q_{t,j}(S_t, \theta^*) \right]$$

$$\leq \sum_{t=1}^T \left[ \sum_{j \in S_t} \tilde{q}_{t,j}(S_t, \tilde{\theta}_t) - \sum_{j \in S_t} \tilde{q}_{t,j}(S_t, \tilde{\theta}^*) \right].$$

Now, define $Q : \mathbb{R}^K \longrightarrow \mathbb{R}$, such that for all $u = \{u_1 \ldots, u_K\} \in \mathbb{R}^K$, $Q(u) = \sum_{i=1}^K \frac{e^{u_i}}{1 + \sum_{j=1}^K e^{u_j}}$. Noting that $S_t$ always contain $K$ elements, we write $S_t = \{i_1, \ldots, i_K\}$ where for all $j, i_j \in [N]$. Finally, for all $t \in [T]$, we let $u_t = (x_{t,i_1}^\top \theta_{t,1}, \ldots, x_{t,i_K}^T \theta_{t,K})^T$ and $u_t^* = (x_{t,i_1}^\top \theta^*, ..., x_{t,i_K}^\top \theta^*)^T$.

We obtain, by a second order Taylor expansion for all $t \geq 1$, that for some convex combination $\bar{u}_t$ of $u_t$ and $u_t^*$, we have:

$$\sum_{t=1}^T \left[ \sum_{j \in S_t} \tilde{q}_{t,j}(S_t, \tilde{\theta}_t) - \sum_{j \in S_t} \tilde{q}_{t,j}(S_t, \tilde{\theta}^*) \right]$$

$$= \sum_{t=1}^T Q(u_t) - Q(u_t^*)$$

$$= \sum_{t=1}^T \nabla Q(u_t^*)^T (u_t - u_t^*) + \frac{1}{2} \sum_{t=1}^T (u_t - u_T^*)^\top \nabla^2 Q_t(\bar{u}_t)(u_t - u_T^*)$$

$$= \sum_{t=1}^T \nabla Q(u_t^*)^T (u_t - u_t^*) + R_2(T), \tag{28}$$

where $R_2(T)$ is a second order term that we will explicit later. Now,

$$\sum_{t=1}^T \nabla Q(u_t^*)^T (u_t - u_t^*)$$

$$= \sum_{t=1}^T \left[ \frac{\sum_{j \in S_t} e^{x_{t,j}^\top \theta^*}(u_j - u_j^*)}{1 + \sum_{j \in S_t} e^{x_{t,j}^\top \theta^*}} - \frac{\sum_{j \in S_t}(e^{x_{t,j}^\top \theta^*} \sum_{i \in S_t} e^{x_{t,i}^\top \theta^*}(u_j - u_j^*))}{\left(1 + \sum_{j \in S_t} e^{x_{t,j}^\top \theta^*}\right)^2} \right]$$

$$= \sum_{t=1}^T \left[ \sum_{j \in S_t} q_{t,j}(S_t, \theta^*) x_{t,j}^\top (\theta_{t,j} - \theta^*) - \sum_{j \in S_t} \sum_{i \in S_t} q_{t,j}(S_t, \theta^*) q_{t,i}(S_t, \theta^*) x_{t,i}^\top (\theta_{t,i} - \theta^*) \right]$$

$$= \sum_{t=1}^T \left[ \sum_{j \in S_t} q_{t,j}(S_t, \theta^*) \left(1 - \sum_{i \in S_t} q_{t,i}(S_t, \theta^*)\right) x_{t,j}^\top (\theta_{t,j} - \theta^*) \right]$$

$$= \sum_{t=1}^T \sum_{j \in S_t} q_{t,j}(S_t, \theta^*) q_{t,0}(S_t, \theta^*) x_{t,j}^\top (\theta_{t,j} - \theta^*)$$

$$\leq \sum_{t=1}^{T} \sum_{j \in S_t} q_{t,j}(S_t, \theta^*) q_{t,0}(S_t, \theta^*) ||x_{t,j}||_{H_t(\theta^*)^{-1}} ||\theta_{t,j} - \theta^*||_{H_t(\theta*)}$$

$$\overset{(a)}{\leq} (1 + \sqrt{6K}W) \sum_{t=1}^{T} \gamma_t(\delta) \sum_{j \in S_t} q_{t,j}(S_t, \theta^*) q_{t,0}(S_t, \theta^*) ||x_{t,j}||_{H_t(\theta^*)^{-1}}$$

$$\overset{(b)}{\leq} (1 + \sqrt{6K}W) \bar{\gamma}_T(\delta) \sqrt{\sum_{t=1}^{T} \sum_{j \in S_t} q_{t,j}(S_t, \theta^*) q_{t,0}(S_t, \theta^*)}$$

$$\cdot \sqrt{\sum_{t=1}^{T} \sum_{j \in S_t} q_{t,j}(S_t, \theta^*) q_{t,0}(S_t, \theta^*) ||x_{t,j}||_{H_t(\theta^*)^{-1}}^2}, \quad (29)$$

where $\bar{\gamma}_T(\delta) := \max_{t \in [T]} \gamma_t(\delta)$. Since $\tilde{\theta}_{t,j} \in C_t(\delta)$ for all $t \geq 1$, inequality (a) is a consequence of Lemma C.4 and the assumption that $A_\delta$ is satisfied. Inequality (b) results from the Cauchy-Schwartz inequality.

Noting that, since $\delta = \frac{1}{K^2 T^2}$, we have that for some constant $C$ which depends only polynomially on W and does not depend on $T, d, K$, $\bar{\gamma}_T(\delta) \leq C\sqrt{d \log(KT)}$. Thus, by combining Lemmas C.1 and C.2 with inequality (29), we get that for some constant $C_1$ which depends only polynomially on W and does not depend on $T, d, K$:

$$\sum_{t=1}^{T} \nabla Q(u_t^*)^T (u_t - u_t^*) \leq C_1 K d \log(KT) \left( \sqrt{\sum_{t=1}^{T} \kappa_{2,t}^* + R(T)} \right)$$

$$\leq C_1 K d \log(KT) \left( \sqrt{\sum_{t=1}^{T} \kappa_{2,t}^*} + \sqrt{R(T)} \right) \quad (30)$$

We now provide a crude upper bound on the second order term $R_2(T)$.

$$R_2(T) = \frac{1}{2} \sum_{t=1}^{T} (u_t - u_T^*)^\top \nabla^2 Q(\bar{u}_t)(u_t - u_T^*)$$

$$\leq \frac{5}{2} \sum_{t=1}^{T} \sum_{j=1}^{K} \sum_{i=1}^{K} x_{t,j}^\top (\tilde{\theta}_{t,j} - \theta^*) x_{t,i}^\top (\tilde{\theta}_{t,i} - \theta^*)$$

$$\leq \frac{5}{2} \sum_{t=1}^{T} \frac{1}{2} \sum_{j=1}^{K} \sum_{i=1}^{K} [(x_{t,j}^\top (\tilde{\theta}_{t,j} - \theta^*))^2 + (x_{t,i}^\top (\tilde{\theta}_{t,i} - \theta^*))^2]$$

$$= \frac{5}{2} K \sum_{t=1}^{T} \sum_{j=1}^{K} (x_{t,j}^\top (\tilde{\theta}_{t,j} - \theta^*))^2$$

$$\leq \frac{5}{2} K \sum_{t=1}^{T} \sum_{j=1}^{K} ||x_{t,j}||_{H_t(\theta^*)^{-1}}^2 ||\tilde{\theta}_{t,j} - \theta^*||_{H_t(\theta^*)}^2$$

$$\leq \frac{5}{2} K \bar{\gamma}_T(\delta)^2 (1 + \sqrt{6K}W)^2 2d(K + \frac{1}{\kappa_2}) \log \left( \lambda_{T+1} + \frac{2TK}{d} \right) \quad (31)$$

where the first inequality results from Lemma C.3 and the last one from Lemmas C.1 and C.4 and the fact that $\tilde{\theta}_{t,j} \in C_t(\delta)$ for all $j \in [K]$.

Using again that $\bar{\gamma}_T(\delta) \leq C\sqrt{d \log(KT)}$, we obtain that for some constant $C_2$ which depends only polynomially on $W$ and does not depend on $T, d, K$:

$$R_2(T) \leq \frac{C_2 d^2 K^3}{\kappa_2} \log(KT)^2$$

Coming back to equation (28) and using the upper bounds given by (30) and (31), we obtain:

$$R(T) - C_1 K d \log(KT)\sqrt{R(T)} \leq \frac{C_2 d^2 K^3}{\kappa_2} \log(KT)^2 + C_1 K d \log(KT) \left( \sqrt{\sum_{t=1}^{T} \kappa_{2,t}^*} \right).$$

Consider the two following cases:

- $C_1 K d \log(KT)\sqrt{R(T)} \leq \frac{R(T)}{2}$.
  Then $R(T) \leq 2\left( \frac{C_2 d^2 K^3}{\kappa_2} \log(KT)^2 + C_1 K d \log(KT) \left( \sqrt{\sum_{t=1}^{T} \kappa_{2,t}^*} \right) \right)$

- Otherwise, $C_1 K d \log(KT) \geq \frac{\sqrt{R(T)}}{2}$, hence $R(T) \leq 4K^2 C_1^2 d^2 \log(KT)^2$.

Hence,

$$R(T) \leq \max\left\{ \frac{2C_2 d^2 K^3}{\kappa_2} \log(KT)^2 + 2C_1 K d \log(KT) \left( \sqrt{\sum_{t=1}^{T} \kappa_{2,t}^*} \right), 4K^2 C_1^2 d^2 \log(KT)^2 \right\}.$$

To finish the proof, we consider the case where $A_\delta$ is not satisfied. In this case, $R(T)$ is still upper bounded by $KT$.

Hence, using that $\delta = \frac{1}{K^2 T^2}$ and by using the law of total probabilities, we conclude that there are some constants $\tilde{C}_1, \tilde{C}_2$ which depends only polynomially on $W$ and do not depend on $T, d, K$ such that:

$$R(T) \leq \tilde{C}_1 K d \log(KT) \left( \sqrt{\sum_{t=1}^{T} \kappa_{2,t}^*} \right) + \frac{\tilde{C}_2 d^2 K^3}{\kappa_2} \log(KT)^2.$$

$\square$

### C.2 Proofs of the main lemmas

**Proof of Lemma C.1.** The proof is similar in spirit to the proof of Lemma B.8 and once again is inspired by the proof of the elliptical potential lemma in [1], while incorporating the local curvature information given by the terms $\{q_{t,j}(S_t, \theta^*) q_{t,0}(S_t, \theta^*)\}$.

$$H_t(\theta^*) = H_{t-1}(\theta^*) + \sum_{i \in S_t} q_{t,i}(S_t, \theta^*) x_{t,i} x_{t,i}^T$$

$$- \sum_{i \in S_t} \sum_{j \in S_t} q_{t,i}(S_t, \theta^*) q_{t,j}(S_t, \theta^*) x_{t,i} x_{t,j}^T + (\lambda_t - \lambda_{t-1}) I_d$$

$$\succeq H_{t-1}(\theta^*) + \sum_{i \in S_t} q_{t,i}(S_t, \theta^*) q_{t,0}(S_t, \theta^*) x_{t,i} x_{t,i}^T, \tag{32}$$

where we used in the last inequality that $\lambda_t \geq \lambda_{t-1}$ and a similar argument as used before. Hence we obtain that

$$\det(H_t(\theta^*)) = \det(H_{t-1}(\theta^*)) \left( 1 + \sum_{i \in S_t} q_{t,i}(S_t, \theta^*) q_{t,0}(S_t, \theta^*) \|x_{t,i}\|_{H_t(\theta^*)^{-1}}^2 \right)$$

Taking the log on both sides and summing from $t = 1$ to $T$, we get:

$$\sum_{t=1}^{T} \log \left( 1 + \sum_{i \in S_t} q_{t,i}(S_t, \theta^*) q_{t,0}(S_t, \theta^*) \|x_{t,i}\|^2_{H_t(\theta^*)^{-1}} \right)$$

$$\leq \sum_{t=1}^{T} \log(\det(H_t(\theta^*))) - \log(\det(H_{t-1}(\theta^*)))$$

$$= \log \left( \frac{\det(H_{T+1}(\theta^*))}{\det(H_1(\theta^*))} \right)$$

$$= \log(\det(H_{T+1}(\theta^*))) \qquad\qquad (\lambda_1 = 1))$$

$$\leq \log \left( \frac{(\mathrm{trace}(H_{T+1}))^d}{d} \right) \qquad \text{(determinant-trace inequality (see [10]))}$$

$$\leq d \log \left( \lambda_{T+1} + \frac{2TK}{d} \right). \qquad\qquad \text{(similarly as in C.1)}$$

Since

$$\sum_{i \in S_t} q_{t,i}(S_t, \theta^*) q_{t,0}(S_t, \theta^*) \|x_{t,i}\|^2_{H_t(\theta^*)^{-1}}$$

$$\leq \frac{1}{\lambda_{min}(H_t(\theta^*))} \sum_{i \in S_t} q_{t,i}(S_t, \theta^*) q_{t,0}(S_t, \theta^*) \|x_{t,i}\|^2_2 \leq \frac{K}{\lambda_1} = K,$$

we get:

$$\sum_{t=1}^{T} \log \left( 1 + \sum_{i \in S_t} q_{t,i}(S_t, \theta^*) q_{t,0}(S_t, \theta^*) \|x_{t,i}\|^2_{H_t(\theta^*)^{-1}} \right)$$

$$\geq \sum_{t=1}^{T} \log \left( 1 + \frac{1}{K} \sum_{i \in S_t} q_{t,i}(S_t, \theta^*) q_{t,0}(S_t, \theta^*) \|x_{t,i}\|^2_{H_t(\theta^*)^{-1}} \right)$$

$$\geq \sum_{t=1}^{T} \frac{1}{2K} \sum_{i \in S_t} q_{t,i}(S_t, \theta^*) q_{t,0}(S_t, \theta^*) \|x_{t,i}\|^2_{H_t(\theta^*)^{-1}} \qquad (\log(1 + x) \geq \tfrac{x}{2} \text{ for } x \in [0, 1])$$

We deduce:

$$\sum_{t=1}^{T} \sum_{i \in S_t} q_{t,i}(S_t, \theta^*) q_{t,0}(S_t, \theta^*) \|x_{t,i}\|^2_{H_t(\theta^*)^{-1}} \leq 2dK \log \left( \lambda_{T+1} + \frac{2TK}{d} \right).$$

To show the second inequality, we come back to equation (32) and further lower bound it using the definition of $\kappa_2$:

$$H_t(\theta^*) \succeq H_{t-1}(\theta^*) + \kappa_2 \sum_{i \in S_t} x_{t,i} x_{t,i}^T.$$

We then conclude on the same way:

$$d \log \left( \lambda_{T+1} + \frac{2TK}{d} \right) \geq \sum_{t=1}^{T} \log \left( 1 + \sum_{i \in S_t} \kappa_2 \|x_{t,i}\|^2_{H_t(\theta^*)^{-1}} \right)$$

$$\geq \sum_{t=1}^{T} \log \left( 1 + \frac{1}{\max(1, K\kappa_2)} \sum_{i \in S_t} \kappa_2 \|x_{t,i}\|^2_{H_t(\theta^*)^{-1}} \right)$$

$$\geq \frac{1}{2(1 + K\kappa_2)} \sum_{i \in S_t} \kappa_2 \|x_{t,i}\|^2_{H_t(\theta^*)^{-1}}$$

Hence,

$$\sum_{i \in S_t} \|x_{t,i}\|^2_{H_t(\theta^*)^{-1}} \le \frac{1}{\kappa_2} 2d(1 + \kappa_2 K) \log\left(\lambda_{T+1} + \frac{2TK}{d}\right)$$

$\square$

**Proof of Lemma C.2.** Since $S_t^*$ and $S_t$ both contain $K$ elements, we write $S_t = \{i_1, ..., i_K\}$, $S_t^* = \{j_1, ..., j_K\}$, and we define $u_t := (x_{t,i_1}^\top \theta^*, ..., x_{t,i_K}^\top \theta^*)^T$, $u_t^* := (x_{t,j_1}^\top \theta^*, ..., x_{t,j_K}^\top \theta^*)^T$ the vectors of the true utilities from products in $S_t$ and $S_t^*$, respectively.

Without loss of generality, we assume that the elements of $u_t$ and $u_t^*$ are sorted by ascending order. Since $S_t^*$ contains the products with the K top utilities, we thus have that for all $i \in [k_t]$, $u_{ti}^* \ge u_{ti}$.

Now, let:

$$g(u) := \sum_{j=1}^{K} \frac{e^{u_j}}{(1 + \sum_{j=1}^{N} e^{u_j})^2}$$

Note that $g(u_t) = \sum_{j \in S_t} q_{t,j}(S_t, \theta^*) q_{t,0}(S_t, \theta^*)$.

Using the mean value theorem, we obtain that:

$$g(u_t) \tag{33}$$

$$= g(u_t^*) + \int_0^1 \nabla g(u_t^* + z(u_t - u_t^*)) dz^\top (u_t - u_t^*)$$

$$= g(u_t^*) + \sum_{i=1}^{K} \int_0^1 \frac{e^{u_{ti}^* + z(u_{ti} - u_{ti}^*)}}{(1 + \sum_{j=1}^{K} e^{u_{tj}^* + z(u_{tj} - u_{tj}^*)})^2} \left(1 - 2\sum_{k=1}^{K} \frac{e^{u_{tk}^* + z(u_{tk} - u_{tk}^*)}}{1 + \sum_{j=1}^{K} e^{u_{tj}^* + z(u_{tj} - u_{tj}^*)}}\right)(u_{ti} - u_{ti}^*) dz$$

$$\le g(u_t^*) + \sum_{i=1}^{K} \left| \int_0^1 \frac{e^{u_{ti}^* + z(u_{ti} - u_{ti}^*)}}{(1 + \sum_{j=1}^{K} e^{u_{tj}^* + z(u_{tj} - u_{tj}^*)})^2} \left(1 - 2\sum_{k=1}^{K} \frac{e^{u_{tk}^* + z(u_{tk} - u_{tk}^*)}}{1 + \sum_{j=1}^{K} e^{u_{tj}^* + z(u_{tj} - u_{tj}^*)}}\right)(u_{ti} - u_{ti}^*) dz \right|$$

$$\overset{(a)}{\le} g(u_t^*) + \sum_{i=1}^{K} \int_0^1 \left| \frac{e^{u_{ti}^* + z(u_{ti} - u_{ti}^*)}}{(1 + \sum_{j=1}^{K} e^{u_{tj}^* + z(u_{tj} - u_{tj}^*)})^2} \left(1 - 2\sum_{k=1}^{K} \frac{e^{u_{tk}^* + z(u_{tk} - u_{tk}^*)}}{1 + \sum_{j=1}^{K} e^{u_{tj}^* + z(u_{tj} - u_{tj}^*)}}\right) \right| dz(u_{ti}^* - u_{ti})$$

$$\overset{(b)}{\le} g(u_t^*) + \sum_{i=1}^{K} \int_0^1 \frac{e^{u_{ti}^* + z(u_{ti} - u_{ti}^*)}}{(1 + \sum_{j=1}^{K} e^{u_{tj}^* + z(u_{tj} - u_{tj}^*)})^2} dz(u_{ti}^* - u_{ti})$$

$$\overset{(c)}{=} g(u_t^*) + \sum_{i=1}^{K} \int_0^1 \frac{e^{u_{ti} + z(u_{ti}^* - u_{ti})}}{(1 + \sum_{j=1}^{K} e^{u_{tj} + z(u_{tj}^* - u_{tj})})^2} dz(u_{ti}^* - u_{ti}), \tag{34}$$

where inequality (a) comes from the fact that $u_{ti}^* \ge u_{ti}$ for all i, (b) uses the inequality $\left|1 - 2\sum_{k=1}^{K} \frac{e^{u_{tk}^* + z(u_{tk} - u_{tk}^*)}}{1 + \sum_{j=1}^{K} e^{u_{tj}^* + z(u_{tj} - u_{tj}^*)}}\right| \le 1$, and equality (c) comes from a change of variable.

We will now link this last term to the regret at time t. For $u \in \mathbb{R}^K$, recall the definition:

$$Q(u) := \sum_{i=1}^{K} \frac{e^{u_i}}{1 + \sum_{j=1}^{K} e^{u_j}}$$

We can express the regret as follows:

$$R(T) = \sum_{t=1}^{T} Q(u_t^*) - Q(u_t)$$

By doing a Taylor expansion at $u_t^*$:

$$R(T) = \sum_{t=1}^{T} \int_0^1 \nabla Q(u_t + z(u_t^* - u_t)) dz^\top (u_t^* - u_t)$$

$$= \sum_{t=1}^{T} \sum_{i=1}^{K} \int_0^1 \frac{e^{u_{ti} + z(u_{ti}^* - u_{ti})}}{1 + \sum_{j=1}^{K} e^{u_{tj} + z(u_{tj}^* - u_{tj})}} \left( 1 - \sum_{k=1}^{K} \frac{e^{u_{tk} + z(u_{tk}^* - u_{tk})}}{1 + \sum_{j=1}^{K} e^{u_{tj} + z(u_{tj}^* - u_{tj})}} \right) dz (u_{ti}^* - u_{ti})$$

$$= \sum_{i=1}^{K} \int_0^1 \frac{e^{u_{ti} + z(u_{ti}^* - u_{ti})}}{(1 + \sum_{j=1}^{K} e^{u_{tj} + z(u_{tj}^* - u_{tj})})^2} dz (u_{ti}^* - u_{ti}).$$

Noting the correspondence of this last term with the second term of (34), we can complete the proof of the lemma as follows:

$$\sum_{t=1}^{T} \sum_{j \in S_t} q_{t,j}(S_t, \theta^*) q_{t,0}(S_t, \theta^*) = \sum_{t=1}^{T} g(u_t)$$

$$\leq \sum_{t=1}^{T} g(u_t^*) + \sum_{t=1}^{T} \sum_{i=1}^{K} \int_0^1 \frac{e^{u_{ti}^* + z(u_{ti} - u_{ti}^*)}}{(1 + \sum_{j=1}^{K} e^{u_{tj}^* + z(u_{tj} - u_{tj}^*)})^2} dz (u_{ti}^* - u_{ti})$$

$$= \sum_{t=1}^{T} g(u_t^*) + R(T)$$

$$= \sum_{t=1}^{T} \sum_{j \in S_t^*} q_{t,j}(S_t^*, \theta^*) q_{t,0}(S_t^*, \theta^*) + R(T)$$

$$= \sum_{t=1}^{T} \kappa_{2,t}^* + R(T). \qquad \text{(by definition of } \kappa_{2,t}^*\text{)}$$

$$\square$$

**Proof of Lemma C.3.** Let $i, k \in [K]$. We first write:

$$\frac{\partial Q}{\partial i} = \frac{e^{u_i}}{1 + \sum_{j=1}^{K} e^{u_j}} - \frac{e^{u_i}(\sum_{j=1}^{K} e^{u_j})}{(1 + \sum_{j=1}^{K} e^{u_j})^2}.$$

Then,

$$\frac{\partial^2 Q}{\partial i \partial k} = -\frac{e^{u_i} e^{u_k}}{(1 + \sum_{j=1}^{K} e^{u_j})^2}$$

$$- \frac{[e^{u_i} e^{u_k} + \mathbf{1}_{i=k} e^{u_i} \sum_{j=1}^{K} e^{u_j}](1 + \sum_{j=1}^{K} e^{u_j})^2 - e^{u_i}(\sum_{j=1}^{K} e^{u_j})2 e^{u_k}(1 + \sum_{j=1}^{K} e^{u_j})}{(1 + \sum_{j=1}^{K} e^{u_j})^4}$$

$$= -\frac{e^{u_i} e^{u_k}}{(1 + \sum_{j=1}^{K} e^{u_j})^2} - \frac{e^{u_i} e^{u_k} + \mathbf{1}_{i=k} e^{u_i} \sum_{j=1}^{K} e^{u_j}}{(1 + \sum_{j=1}^{K} e^{u_j})^2} + \frac{2 e^{u_i}(\sum_{j=1}^{K} e^{u_j}) e^{u_k}}{(1 + \sum_{j=1}^{K} e^{u_j})^3}.$$

Thus,

$$\left| \frac{\partial^2 Q}{\partial i \partial k} \right| \leq \left| \frac{e^{u_i} e^{u_k}}{(1 + \sum_{j=1}^{K} e^{u_j})^2} \right| + \left| \frac{e^{u_i} e^{u_k}}{(1 + \sum_{j=1}^{K} e^{u_j})^2} \right| + \left| \frac{e^{u_i} \sum_{j=1}^{K} e^{u_j}}{(1 + \sum_{j=1}^{K} e^{u_j})^2} \right| + 2 \left| \frac{e^{u_i}(\sum_{j=1}^{K} e^{u_j}) e^{u_k}}{(1 + \sum_{j=1}^{K} e^{u_j})^3} \right|$$

$$\leq 5.$$

**Proof of Lemma C.4.**

By the multivariate mean value theorem:

$$g_t(\theta_1) - g_t(\theta_2) = \nabla \mathcal{L}_t^{\lambda_t}(\theta_2) - \nabla \mathcal{L}_t^{\lambda_t}(\theta_1)$$

$$= \int_0^1 \nabla^2 \mathcal{L}_t^{\lambda_t}(\theta_1 + z(\theta_2 - \theta_1)) dz (\theta_2 - \theta_1)$$

Hence
$$\|g_t(\theta_1) - g_t(\theta_2)\|_{G_t^{-1}(\theta_1,\theta_2)} = \|\theta_1 - \theta_2\|_{G_t(\theta_1,\theta_2)} \tag{35}$$
where $G_t(\theta_1, \theta_2) := \int_0^1 \nabla^2 \mathcal{L}_t^{\lambda_t}(\theta_1 + z(\theta_2 - \theta_1))dz$.

Using Proposition A.5, we have that:
$$G_t(\theta_1, \theta_2) \succeq \frac{1}{(1+\sqrt{6KW})} H_t(\theta_1) \tag{36}$$

As a result, by combining (36) and (35):
$$\begin{aligned}
\|\theta_1 - \theta_2\|_{H_t(\theta_1)} &\leq (1 + \sqrt{6K}W)^{1/2}\|\theta_1 - \theta_2\|_{G_t(\theta_1,\theta_2)} \\
&= (1 + \sqrt{6K}W)^{1/2}\|g_t(\theta_1) - g_t(\theta_2)\|_{G_t^{-1}(\theta_1,\theta_2)} \\
&\leq (1 + \sqrt{6K}W)\|g_t(\theta_1) - g_t(\theta_2)\|_{H_t^{-1}(\theta_1)}.
\end{aligned}$$

$\square$

## C.3 Construction of the confidence set

In this section, we build upon the new Bernstein-like tail inequality for self-normalized vectorial martingales developed in [17] to derive a confidence set on $\theta^*$.

Remember that:
$$C_t(\delta) := \{\theta \in \Theta \mid \|g_t(\theta) - g_t(\hat{\theta}_t)\|_{H_t^{-1}(\theta)} \leq \gamma_t(\delta)\}$$
Our objective is to prove the following proposition.

*Proposition 3.3 2.* Let $\delta \in (0,1]$. Then $\mathbb{P}(\forall t, \theta^* \in C_t(\delta)) \geq 1 - \delta$.

We start by a few technical considerations and auxiliary lemmas. The proof of this result relies on a Bernstein concentration inequality which is a variant of the following theorem:

**Theorem C.5** (Theorem 4 in [2]). *Let $\{\mathcal{F}_t\}_{t=1}^{\infty}$ be a filtration. Let $\{x_t\}_{t=1}^{\infty}$ be a stochastic process in $\mathcal{B}_2(d)$ such that $x_t$ is $\mathcal{F}_t$ measurable. Let $\{\varepsilon_t\}_{t=2}^{\infty}$ be a martingale difference sequence such that $\varepsilon_t$ is $\mathcal{F}_{t-1}$ measurable. Furthermore, assume that conditionally on $\mathcal{F}_t$ we have $|\varepsilon_t| \leq 1$ almost surely, and note $\sigma_t^2 = \mathbb{E}\left[\varepsilon_t^2 | \mathcal{F}_t\right]$. Let $\{\lambda_t\}_{t=1}^{\infty}$ be a predictable sequence of non-negative scalars. For any $t \geq 1$ define:*
$$H_t = \sum_{s=1}^{t-1} \sigma_s^2 x_s x_s^T + \lambda_t I_d, \qquad S_t = \sum_{s=1}^{t-1} \varepsilon_s x_s.$$

*Then for any $\delta \in (0,1]$:*
$$\mathbb{P}\left(\exists t \geq 1, \|S_t\|_{H_t^{-1}} \geq \frac{\sqrt{\lambda_t}}{2} + \frac{2}{\sqrt{\lambda_t}} \log\left(\frac{2^d \det\left(H_t\right)^{\frac{1}{2}} \lambda^{-\frac{d}{2}}}{\delta}\right)\right) \leq \delta.$$

The above Bernstein inequality is of the same flavor as Theorem 1 in [1], but is taking into account information on the local curvature of the reward function.

In our setting, we consider:
$$H_t = \sum_{s=1}^{t-1}\left[\sum_{i \in S_s} q_{s,i}(S_s, \theta^*)x_{s,i}x_{s,i}^T - \sum_{i \in S_s}\sum_{j \in S_s} q_{s,i}(S_s, \theta^*)q_{s,j}(S_s, \theta^*)x_{s,i}x_{s,j}^T\right] + \lambda_t I_d$$

$$U_t = \sum_{s=1}^{t-1}\sum_{j \in S_s} \varepsilon_{s,j}x_{s,j}$$

where $\varepsilon_{s,j} = y_{s,j} - q_{s,j}(S_s, \theta^*)$.

Note that we cannot directly write $H_t, U_t$ under the form required in Theorem C.5 since for all $s$, the variables $\{\varepsilon_{s,j}\}_{j \in S_s}$ are correlated. We show below that we can still prove similar concentration guarantees on $\|U_t\|_{H_t^{-1}}$.

**Theorem C.6.** *for any $\delta \in (0, 1]$:*

$$\mathbb{P}\left(\exists t \geq 1, \|U_t\|_{H_t^{-1}} \geq \frac{\sqrt{\lambda_t}}{4} + \frac{4}{\sqrt{\lambda_t}} \log\left(\frac{2^d \det{(H_t)}^{\frac{1}{2}} \lambda_t^{-\frac{d}{2}}}{\delta}\right)\right) \leq \delta.$$

Note that this expression is almost identical to the one in Theorem C.5 except for some minor constant modification.

The proof follows the same line as the proof of Theorem 4 in [2], but the analysis differs because of the non independence of the variables $\{\varepsilon_{s,j}\}_{j \in S_s}$. In particular, we analyse the behavior of the global variable $z_s := \sum_{j \in S_s} \varepsilon_{s,j} \xi^\top x_{s,j}$.

As in [2], we consider the non regularized hessian

$$\bar{H}_t = \sum_{s=1}^{t-1} \left[\sum_{i \in S_s} q_{s,i}(S_s, \theta^*) x_{s,i} x_{s,i}^T - \sum_{i \in S_s} \sum_{j \in S_s} q_{s,i}(S_s, \theta^*) q_{s,j}(S_s, \theta^*) x_{s,i} x_{s,j}^T\right]$$

and for all $\xi \in B(0, 1/2)$, we let:

$$M_0(\xi) = 1 \qquad \text{and} \qquad M_t(\xi) = exp(\xi U_t - \|\xi\|_{\bar{H}_t}^2).$$

Note that we define only $M_t(\xi)$ for $\xi \in B(0, 1/2)$ whereas $\xi \in B(0, 1)$ in [17] and [2]. In the following, we consider the filtration $\mathcal{F}_t$ engendered by $\{\{\vec{x_s}, \vec{\epsilon_s}\}_{s=1}^{t-1}, \vec{x_t}\}$. The main ingredient of the proof is to show that relatively to $\mathcal{F}_t$, $M_t(\xi)$ is still a super martingale. The rest of the proof follows immediately from [17] and [2].

To bound $\mathbb{E}\left[\exp(\xi^T U_t)|\mathcal{F}_{t-1}\right]$, we first state the following lemma, whose proof can be found in [2]:

**Lemma C.7.** *Let $\varepsilon$ be a centered random variable of variance $\sigma^2$ and such that $|\varepsilon| \leq 1$ almost surely. Then for all $\lambda \in [-1, 1]$:*

$$\mathbb{E}\left[\exp(\lambda \epsilon)\right] \leq 1 + \lambda^2 \sigma^2.$$

**Lemma C.8.** *For all $\xi \in B(0, 1/2)$, $\{M_t(\xi)\}_{t=0}^\infty$ is a nonnegative super martingale.*

*Proof.* Note that for all $s \geq 1$, there is a single index $i \in S_s \cup \{0\}$ for which $y_{s,i} = 1$, and $y_{s,j} = 0$ for all $j \in S_s \cup \{0\} \setminus \{i\}$. Besides, we have $\mathbb{P}(y_{s,i} = 1) = q_{s,i}(S_s, \theta^*)$. Hence, conditional on $\mathcal{F}_s$, the variance of $\xi^\top z_s$ can be expressed as:

$$\sigma^2(\xi^\top z_s|\mathcal{F}_s)$$

$$= \mathbb{E}\left(\left(\sum_{j \in S_s}(y_{s,j} - q_{s,j}(S_s, \theta^*))\xi^\top x_{s,j}\Big|\mathcal{F}_s\right)^2\right) - \left(\mathbb{E}\left(\sum_{j \in S_s}(y_{s,j} - q_{s,j}(S_s, \theta^*))\xi^\top x_{s,j}\Big|\mathcal{F}_s\right)\right)^2$$

$$= \mathbb{E}\left(\left(\sum_{j \in S_s}(y_{s,j} - q_{s,j}(S_s, \theta^*))\xi^\top x_{s,j}\Big|\mathcal{F}_s\right)^2\right)$$

$$= \mathbb{E}\left(\sum_{j \in S_s}\sum_{i \in S_s}(y_{s,j}\xi^\top x_{s,j})(y_{s,i}\xi^\top x_{s,i})\Big|\mathcal{F}_s\right)$$

$$- 2\mathbb{E}\left(\sum_{j \in S_s}y_{s,j}\xi^\top x_{s,j}\Big|\mathcal{F}_s\right) \cdot \left(\sum_{j \in S_s}\xi^\top x_{s,j}q_{s,j}(S_s, \theta^*)\right) + \left(\sum_{j \in S_s}\xi^\top x_{s,j}q_{s,j}(S_s, \theta^*)\right)^2$$

$$= \mathbb{E}\left(\sum_{j \in S_s}y_{s,j}(\xi^\top x_{s,j})^2\Big|\mathcal{F}_s\right) - 2\left(\sum_{j \in S_s}\xi^\top x_{s,j}q_{s,j}(S_s, \theta^*)\right)^2 + \left(\sum_{j \in S_s}\xi^\top x_{s,j}q_{s,j}(S_s, \theta^*)\right)^2$$

$$= \sum_{j \in S_s}(\xi^\top x_{s,j})^2 q_{s,j}(S_s, \theta^*) - \left(\sum_{j \in S_s}\xi^\top x_{s,j}q_{s,j}(S_s, \theta^*)\right)^2. \tag{37}$$

Now, noting that $U_{t-1}$ is $\mathcal{F}_{t-1}$-measurable, we have that for all $t \geq 1$:

$$\mathbb{E}\left[\exp(\xi^T U_t)|\mathcal{F}_{t-1}\right] = \exp(\xi^T U_{t-1})\mathbb{E}\left[\exp(\xi^\top z_{t-1})|\mathcal{F}_{t-1}\right].$$

Let $i \in S_{t-1} \cup \{0\}$ be the index for which $y_{t-1,j} = 1$. We have $y_{t-1,j} = 0$ for all $j \in S_{t-1} \setminus \{i\}$.
If $i \in S_{t-1}$, using that $\|x_{t,j}\| \leq 1$ for all $t, j$ and $\|\xi\| \leq 1/2$, we have the following inequality:

$$|z_{t-1}| \leq (1 - q_{t-1,i}(S_{t-1}, \theta^*))|\xi^\top x_{t-1,i}| + \sum_{j \in S_{t-1} \setminus \{i\}} q_{t-1,j}(S_{t-1}, \theta^*)|\xi^\top x_{t-1,j}|$$

$$\leq \frac{1}{2}\left(1 + \sum_{j \in S_{t-1}} q_{t-1,j}(S_{t-1}, \theta^*)\right)$$

$$\leq 1$$

Otherwise, $i = 0$ and we have that:

$$|z_{t-1}| \leq \sum_{j \in S_{t-1}} q_{t-1,j}(S_{t-1}, \theta^*)|\xi^\top x_{t-1,j}| \leq 1$$

Since $|z_{t-1}| \leq 1$, we can apply Lemma C.7 and obtain:

$$\begin{aligned}
\mathbb{E}\left[\exp(\xi^T U_t)|\mathcal{F}_{t-1}\right] &= \exp(\xi^T U_{t-1})\mathbb{E}\left[\exp(\xi^\top z_{t-1})|\mathcal{F}_{t-1}\right] \\
&\leq \exp(\xi^T U_{t-1})(1 + \sigma^2(\xi^\top z_{t-1}|\mathcal{F}_{t-1})^2) \\
&\leq \exp(\xi^T U_{t-1} + \sigma^2(\xi^\top z_{t-1})^2|\mathcal{F}_{t-1}) \quad\quad (1 + x \leq e^x)
\end{aligned}$$

Noting that from (37), $\sigma^2(\xi^\top z_{t-1}\big|\mathcal{F}_{t-1})^2$ is exactly equal to $\|\xi\|^2_{\bar{H}_t} - \|\xi\|^2_{\bar{H}_{t-1}}$, it leads to:

$$\begin{aligned}
&\mathbb{E}\left[M_t(\xi)|\mathcal{F}_{t-1}\right] \\
&= \mathbb{E}\left[\exp\left(\xi^T U_t - \|\xi\|^2_{\bar{H}_t}\Big|\mathcal{F}_{t-1}\right)\right] \\
&= \mathbb{E}\left[\exp\left(\xi^T U_t\right)|\mathcal{F}_{t-1}\right]\exp\left(-\|\xi\|^2_{\bar{H}_t}\right) \quad\quad (\bar{H}_t \text{ is } \mathcal{F}_{t-1}\text{- measurable}) \\
&\leq \exp\left(\xi^T U_{t-1} + \sigma^2(\xi^\top z_{t-1}|\mathcal{F}_{t-1})^2 - \|\xi\|^2_{\bar{H}_t}\right) \\
&= \exp\left(\xi^T U_{t-1} - \|\xi\|^2_{\bar{H}_{t-1}}\right) \\
&= M_{t-1}(\xi)
\end{aligned}$$

which shows that $\{M_t(\xi)\}_{t=0}^{\infty}$ is a super martingale. $\quad\square$

**Proof of Theorem C.6.** Using that $\{M_t(\xi)\}_{t=0}^{\infty}$ is a super martingale by Lemma C.8, the proof follows the proof of Theorem 4 in [2] and Theorem 1 in [17], with some minor modification since $\xi$ now belongs to $B(0, 1/2)$ instead of $B(0, 1)$ to guarantee that $\{M_t(\xi)\}_{t=0}^{\infty}$ is a super martingale. In the proof of Theorem 1 in [17], for any scalar $\beta$, we now define $h$ to be the density of an isotropic normal distribution of precision $\beta^2$ truncated on $B(0, 1/2)$ (instead of $B(0, 1)$), and $g$ the density of the normal distribution of precision $2H_t$ truncated on the ball $B(0, 1/4)$ (instead of $B(0, 1/2)$). The upper bound on the ratio of the normalisation constants $\frac{N(g)}{N(h)}$ given by Lemma 6 of [17] remains identical, hence following [2], [17] and taking $\xi_0 = \frac{H_t^{-1} U_t}{\|U_t\|_{H_t^{-1}}}\frac{\beta}{4\sqrt{2}}$ instead of $\xi_0 = \frac{H_t^{-1} U_t}{\|U_t\|_{H_t^{-1}}}\frac{\beta}{2\sqrt{2}}$, we finally obtain that:

$$\mathbb{P}\left(\exists t \geq 1, \|U_t\|_{H_t^{-1}} \geq \frac{\sqrt{\lambda_t}}{4} + \frac{4}{\sqrt{\lambda_t}}\log\left(\frac{2^d \det\left(H_t\right)^{\frac{1}{2}}\lambda_t^{-\frac{d}{2}}}{\delta}\right)\right) \leq \delta.$$

$\square$

We are now ready to complete the proof of Proposition 3.3.

**Proof of Proposition 3.3**. Since, $\hat{\theta}_t$ minimizes $\mathcal{L}_t^{\lambda_t}(\theta)$, we have that $\nabla \mathcal{L}_t^{\lambda_t}(\hat{\theta}_t) = 0$. Hence,

$$g_t(\hat{\theta}_t) = \sum_{s=1}^{t-1} \sum_{j \in S_s} q_{s,j}(S_s, \theta) x_{t,j} + \lambda_t \hat{\theta}_t = \sum_{s=1}^{t-1} \sum_{j \in S_s} y_{s,j} x_{s,j}$$

As a result,

$$g_t(\hat{\theta}_t) - g_t(\theta^*) = \sum_{s=1}^{t-1} \sum_{j \in S_s} (y_{s,j} - q_{s,j}(S_s, \theta^*)) x_{s,j} - \lambda_t \theta^*$$
$$= U_t - \lambda_t \theta^*.$$

Therefore, since $\|\theta^*\| \leq W$ and $H_t(\theta^*)^{-1} \preceq \frac{1}{\lambda_t} I_d$, we get

$$\|g_t(\hat{\theta}_t) - g_t(\theta^*)\|_{H_t(\theta^*)^{-1}} \leq \|U_t\|_{H_t(\theta^*)^{-1}} + \sqrt{\lambda_t} W.$$

The proof concludes with a straightforward application of Theorem C.6, combined with the following upper bound on $\det(H_t)$ resulting from the determinant-trace inequality:

$$\det(H_t) \leq \left( \frac{\text{trace}(H_t)}{d} \right)^d \leq \left( \lambda_t + \frac{2tK}{d} \right)^d.$$

## D   Numerical experiments - Comparison to the ONSP policy from [42]

In this section, we numerically compare the performance of our ONS based pricing policy for self concordant functions (ONSSC) and the ONSP policy from [42] when only a single product needs to be priced and the price sensitivity is unitary (which is the setting of ([42]), and the noise has a logistic distribution.

We study the performance of these two algorithms for different values of $W, d$ and different distributions of the contexts $\{x_t\}$. The optimal parameter $\theta^*$ is set as $\theta^* = Y \times Z/\|Z\|$ where $Y \sim \mathcal{U}([0, W])$ and $Z$ is sampled from a multivariate Gaussian distribution $\mathcal{N}(0, I_d)$. In the two first set of experiments, we assume that the contexts $\{x_t\}$ are generated independently at each period according to a multivariate Gaussian distribution $\mathcal{N}(0, I_d)$, then renormalized so that $\|x_t\| = 1$. In the third set of experiments, we assume that the product feature vectors $\{x_t\}$ are generated independently at each period according to a multivariate exponential distribution with scale parameter $\beta = 1$, then renormalized so that $\|x_t\| = 1$. In the fourth and last set of experiments, we consider adversarial contexts constructed similarly as in [42]: we set $d = 2$ and we divide the time horizon into $\log(T)$ epochs, such that each epoch $\mathcal{E}_t$ is constituted of time steps $\{2^{k-1}, \ldots, 2^k - 1\}$. For all $t \in \mathcal{E}_k$, we then set $x_t = [0, 1]^T$ if $k \equiv 0$ [2] and $x_t = [1, 0]^T$ if $k \equiv 1$ [2].

The results of our experiments are displayed on Figure 1. We compare the cumulative regret obtained by the two algorithms for $T = 10^4$ steps. As $W$ grows, the parameter $\gamma$ in the policy from [42] becomes exponentially small. We observe that in this case, our policy achieves a significantly better regret in all sets of experiments. This experimentally supports our claim that avoiding to explicitly use $\kappa$ in the descent step may lead to more practical algorithms.

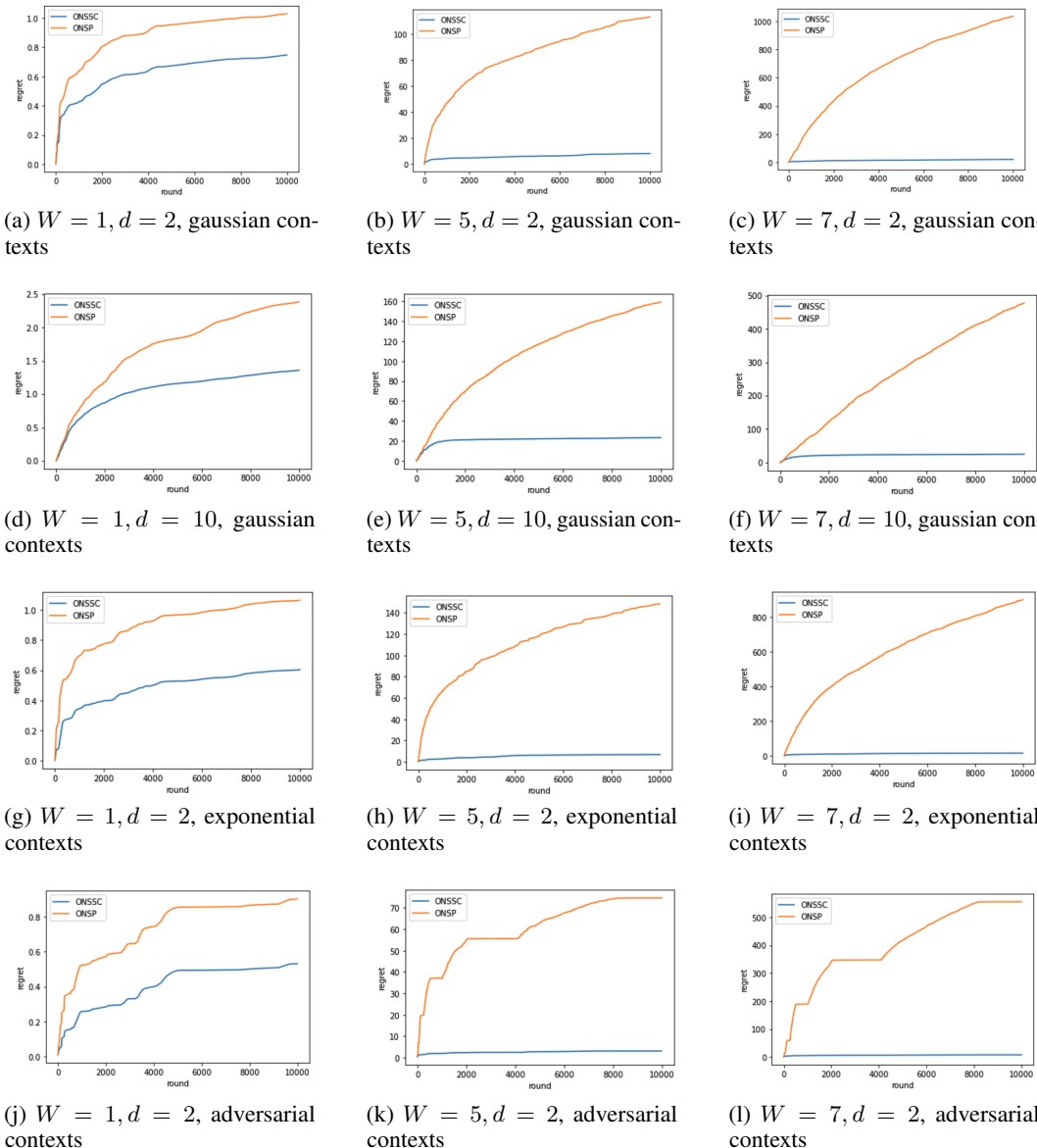

Figure 1: Cumulative regret of the ONSSC and the ONSP policies for different values of $W, d$ and different distributions of the contexts $\{x_t\}$.