# OpenReview forum: "Dynamic pricing and assortment under a contextual MNL demand"
_NeurIPS.cc/2022/Conference — NeurIPS 2022 Accept_

### Official Review · Reviewer_rfSu · 2022-07-07

**Rating:** 6
**Confidence:** 4
**Soundness:** 4 excellent
**Presentation:** 2 fair
**Contribution:** 3 good

**Summary:**

In this work, the authors study two different online-learing problems, a multi-product dynamic pricing problem and a multi-product dynamic assortment optimization problems, under a multinomial logit (MNL) model choice model. In the pricing problem, at each round t, a customer comes with a candidate set of features describing her preference on corresponding products, and the seller (we) are asked to set prices for these candidate products. After seeing these prices, the customer will choose at most one of them to buy, according to the MNL model with linear parameters. The goal is to maximize the expected revenue. In the assorting problem, the situation is slightly different: At each round t, a customer comes with a set of features describing her preference on all products, and the seller (we) are asked to propose a list of $\leq K$ products as candidates. After seeing these candidates, the customer would take at most one product from these candidates. In the assorting problem, the goal is to maximize the probability of not-buying-empty.

In this work, the authors propose algorithms for both problems: For the pricing prolem, they propose an Online-Newton (ON)-based pricing algorithm which achieves an $O(d\log{T}\sqrt{T})$ regret. This result matches the existing information-theoretic lower bound at $\tilde{O}(\sqrt{T})$. For the assorting problem, they propose an OFU-MNL algorithm that achieves an $O(d\sqrt{\kappa T}+d^2\frac{\log(T)}{\kappa})$ regret. This result improve the existing results by reducing the dependence on $\frac1{\kappa}$ as $\kappa=o(1)$.

**Questions:**

Please point it out directly in the rebuttal if I am wrong in the following questions/suggestions.

1, In Line 34 the authors mentioned the regret on the assorting problem, without firstly describing the goal of assorting and the (informal) definition of its regret. This could be confusing to the readers.

2, In Line 86, the authors mentioned that an $O(d\log{T})$ regret is implied by their result for single-product pricing problem. However, I did not find such a reduction in the paper. Is there a remark/corollary regarding this, or a trivial reduction to this result?

3, In Line 93, the authors mentioned that their ON-based pricing algorithm outperforms the ONS-based pricing algorithm in [41] by improving the dependence on a "potentially exponentially small constant". However, I did not find (1) how the small constant is like and (2) what improvement this work has made, in either the main pages or Appendix D. Only descriptive words are in Line 181-191

I guess this small constant could be $\kappa$ as is mentioned in Line 190-191, but I am not quite sure: (1) a $\kappa$ is defined for the assorting problem (see Line 267) instead of the pricing problem (as they have different definitions on the $q_{t,j}$ functions), and (2) the constant in the regret of Algorithm 1 is also $\Omega(\frac1{\kappa^2}$ (see Line 592). Please explain this issue briefly as [41] adopts a quite similar method and this work indeed outperforms [41] in practice.

4, In Line 139, I think it is not a proper way to describe the probabilistic distribution like this: As the authors only specified the probability of product $j$ being purchased, a straightforward question could be: how are these purchases correlated to each other? Of course, we know from context that these purchases are mutually exclusive and the probability is a multinomial on these products, but this could still be very misleading. I suggest the authors clarify this part by saying "the customer purchases one product $j\in[k_t]\cup 0$ ...".

5, There should be an explanation on Assumption 2.3. My understanding: price elasticity (or "price sensitivity" in this work) should not be too small (i.e., close to 0). This is reasonable and practical, but needs explanation.

6, In Line 166, where does this $\lambda$ sequence come from? This makes the adjacent paragraph (Line 166-169) unclear. (A minor issue: for the convexity of $l_t$, either prove it or cite it.)

7, In Line 196, the author said that they treated $K$ as a constant. This is important for regret analysis and comparison, so I suggest the authors to make it as an assumption and state it with other assumptions (e.g., after Assumption 2.3).

8, In Line 219, the author said "it is meaningful only with random price experimentation". This seems not very clear to me. Does it mean that you have to further make use of the randomness of $p_{t,j}$ to upper bound the first term of (1)?

Maybe an equation showing how to apply Lemma 2.6 to the proof of Theorem 2.4 would be more informative and concise.

9, In Line 257, the author denoted the "optimal assortment" as $S_t^*$. My understanding: $S^* = \arg\max_S q_{t,j}(S, \theta^*)$, i.e., minimize the prob of choosing nothing. This is equivalent to maximizing the sum of $exp\{x^{\top}\theta\}$? Is that right?

Besides, I noticed a number of notation inconsistence in this paper. Therefore, I strongly recommend the authors to carefully examine the paper and fix all notation issues.

**Limitations:**

The authors indeed discussed the potential extension of this work (e.g., "the nested logit model" as mentioned in their conclusions). However, there's not much on the social impact. This is not a big issue as it is a theoretic-oriented research, but I recommend the authors to discuss more on the limitations and potential social impacts (especially these negative aspects) in their Appendix.

**Strengths And Weaknesses:**

Strength:

1, This work studies online multi-product pricing and assortment problems and proposes provable algorithms. The problem setting is meaningful. Their regret upper bounds are either optimal (for pricing) or better than existing results (for assortments).

2, For the pricing problem, they use Online Newton method + self-concordant-like property to substitute the role of an ONS algorithm, while also saving the regret dependence on constants. Their numerical result (in Appendix) also shows the significance of their work over existing methods in practice.

Weakness:

1, Their improvement from existing theoretical results on the assorting problem is not substantial.

2, Problem formulation and literature reviews are insufficient: It is unclear how to place this work in the stream of pricing/assorting researches.

Overall, this is a good work to get in, and I would like to raise my score as long as the authors properly address my concerns.

---

> ### Author Response · Authors · 2022-08-02
> **Response to Reviewer rfSu**
>
> Thank you for your very detailed feedback. We would first like to comment on the weaknesses pointed out.
>
> ‘Problem formulation and literature reviews are insufficient: It is unclear how to place this work in the stream of pricing/assorting researches.’ We will extend the literature review section in the revised version and we comment on this now. The objective of this paper was twofold: first, propose a dynamic pricing algorithm for multiple product pricing with adversarial contexts and feature-based price sensitivities under the MNL choice model. This directly extends the setting of [22], with adversarial features instead of features drawn iid from a distribution. We note that with adversarial features, even the single product case with feature-based price sensitivity was not solved. We thus propose a new algorithm that combines an ONS-based update and random price shocks. Even though these two ingredients have already been used separately in the pricing literature (in [41] and [30], respectively), it was not clear that using simultaneously an ONS-based algorithm and random price shocks would lead to an algorithm with near-optimal regret (see the first point in the response to reviewer eXno for additional discussion on this point).
>  Secondly, we aimed to show that under the widely used MNL choice model, the nonlinearity of the customer choice model does not impair the design of good pricing and assortment algorithms, in the sense that the algorithms do not require the knowledge of the ‘degree of non-linearity of the problem’ (captured by the parameter $\kappa$) and that their regret do not scale with $\kappa$ (which we show in theory for the assortment problem and illustrate numerically for the pricing problem). This important issue has been pointed out and solved for contextual logistic bandits (see [2],[17]), but previous contextual assortment algorithms under the MNL model ([31], [32], [12]) suffer from it. It has also been mentioned in the dynamic pricing setting (see Section 7 in [41]).
>
>
>
> We now present our response to the questions and concerns.
>
> * 2) We agree that this is not clearly stated and we will include a complete proof in the Appendix.  First, note that the algorithm used to obtain the $O(d\log(T))$ upper bound in the single product pricing without price sensitivity setting is our Algorithm 1 without the price shocks. If there is no price sensitivity (i.e. $\alpha = \alpha^*=1$) and a single product ($k_t = 1$), note that from Lemma 2.6, we get $\sum x_t^T(\theta_t-\theta_*) = O(d\log(T))$. Combining this with Lemma 2.5, we obtain the desired guarantee.
>
>  * 3) For the pricing problem, we will move the exact definition of $\kappa$ in the main body: $\kappa = \min_{p\in [0,p_{\max}],\gamma\in B(W),j\in \{1,\ldots k_t\}} q_{t,j}(\gamma, p) q_{t,0}(\gamma, p)$. Note that this corresponds to the constant $C_{down}/C_{exp}$ defined in equation (12) in [41] and that the issue of the dependence of the regret in  $C_{down}/C_{exp}$ has been pointed in [41] (see Section 7).
> We will now precise the statement in line 93:
> In [41], $\theta_t$ is updated directly by the traditional Online Newton Step algorithm, whereas we use a different update rule (we use the hessian in the update instead of an approximation $A_t$ of the hessian, and the descent parameter $\mu$ is different). Note that [41] needs to use the exp-concavity parameter in the descent step to obtain their theoretical guarantee. However, because the MNL loss is self-concordant-like, we show that our update rule allows us to achieve near-optimal theoretical regret without using the exp-concavity parameter in the descent step (the proof uses Lemma B.5, which exploits the inequality proved in A.4 using the self-concordant-like property of the loss). Hence, what we show exactly is that we can obtain a near-optimal regret (in d and T) with an algorithm that does not use $\kappa$. We then show that our algorithm numerically outperforms the algorithm in [41] for the single product without price sensitivity when the noise has a logistic distribution. However, the constant $\kappa$ still appears in our theoretical regret bound and it is not clear yet if it can be removed in the analysis.
>
> * 5) We will add an explanation for this assumption. Note that it also appears in [22] (Assumption 2.1).
>
> * 6) The $\{\lambda\}$ are defined as $\lambda_t = d\log(t), \lambda_1 = 1$ (see Theorem 2.4). We will introduce them earlier to make it easier to follow.
>
> * 8) What we meant here is that it is not possible, in general, to derive directly Theorem 2.4 from Lemma 2.6 since the left-hand side of Lemma 2.6 can be zero if all prices posted are 'uninformative'. Hence our use of random price shocks to circumvent this issue.
>
> * 9) Yes, this is right.
>
> Thank you for the additional suggestions in 1), 4) and 7). We will make the changes in the revised version. We will also carefully examine the notations and add in Appendix a section on potential social impacts.

---

> > ### Comment · Reviewer_rfSu · 2022-08-06
> > **Thank the authors for your reply!**
> >
> > The authors' response has answered my questions and made sufficient clarification to most issues I was concern. I tend to agree that the theoretical dependence w.r.t. $1/\kappa$ is an interesting problem for further study. Therefore, I would like to regrade this paper to 6. However, I highly recommend the authors to update their submissions according to these points above before the deadline of discussion session.

---

> > > ### Author Response · Authors · 2022-08-09
> > > **Thanks for your comment!**
> > >
> > > Please find a first revised version of the paper in the supplementary material.

---

### Official Review · Reviewer_zMXk · 2022-07-10

**Rating:** 6
**Confidence:** 2
**Soundness:** 3 good
**Presentation:** 3 good
**Contribution:** 3 good

**Summary:**

In this paper, the authors studied contextual dynamic pricing and assortment optimization problems under the MNL choice model. They proposed two algorithms: (1) a dynamic pricing algorithm which combines a variant of the ONS algorithm and random price shocks, and showed that this algorithm achieves $O(d \sqrt{T} \log{T})$ regret under adversarial arrival. (2) an optimistic algorithm for the adversarial MNL contextual bandits problem, and showed that it achieves better dependency on parameter $\kappa$ than existing algorithms, where $\kappa$ is a problem-dependent constant that measures deviation from the linear model.

**Questions:**

1. In Algorithm 2, there is a step that requires one to choose the set of $K$ items maximizing the expected revenue. Are you simply selecting the $K$ items with the highest $x_{t, i}^\top \tilde{\theta}_i$ here?
2. If so, any ideas what will happen when the rewards are no longer uniform and we also need to consider assortments with size less than $K$?


**Limitations:**

None.

**Strengths And Weaknesses:**

Strengths:
1. The dynamic pricing policy proposed in Section 2 uses interesting ideas that combine ONS method with random price shocks, and is shown to attain near-optimal performance.
2. The optimistic algorithm for adversarial MNL contextual bandits proposed in Section 3 is shown to achieve better dependency on $\kappa$, and does not assume knowledge of $\kappa$ a priori.
3. The authors provide clear and thorough proof for their theoretical results.

Weaknesses:
1. The authors provided a brief discussion of related literature on contextual dynamic pricing/assortment problems in the beginning of section 1. It could be desirable to have a separate and more detailed discussion of related work that compares this work with the previous literature.
2. While the regret bounds are theoretically sound, it could be good to also include some numerical studies here to demonstrate the performance of the algorithm and verify the dependency on parameters $T$ and $d$ of the regret bound. Another thing that is missing from the current work is the runtime for both algorithms.
3. It appears that the design of the algorithms leverages the self-concordant property of the MNL log likelihood function. More discussions can be provided here to motivate why the use of self-concordant property is important to the proof.

---

> ### Author Response · Authors · 2022-08-02
> **Response to Reviewer zMXk**
>
> Thank you for your comments and your suggestion to provide a more detailed literature review section and numerical studies for the final version of the paper. We will also make it clearer how the self-concordant-like property is used. In the pricing setting, we can point you to the proof of  Lemma B.5, which exploits the inequality proved in Appendix A.4., and in the assortment setting, we can point to Lemmas C.1 and C.2. In the revised version, we will provide more intuition about this. We will also add the running times for the two algorithms (for the pricing algorithm, see Question 2 of Reviewer eXno; for the assortment algorithm, note that the current algorithm is mainly of theoretical interest, since it shows that it is possible to achieve a regret whose first order term does not scale with $\kappa$. However, computing each $\tilde{\theta}_{t,j}$ remains computationally expensive).
>
> Questions:
> 1) Yes, we select the $K$ items with highest $x_{t,i}^T\tilde{\theta}_i$.
> 2) This is a very good point: even though the algorithm we propose is still valid when the revenues are non-uniform (this time, by offering the set that maximizes $\tilde{r}(S,\tilde{\theta})$ instead of the $K$ items with highest $x_{t,i}^T\tilde{\theta}_i$), the current analysis would lead to an additional factor $1/\kappa$ in the regret upper bound. It remains an open problem how to remove the dependency on the parameter in this case.

---

> > ### Comment · Reviewer_zMXk · 2022-08-08
> > **Thank you for your response**
> >
> > I would like to thank the authors for putting together the response and addressing my questions above. It would be great if the authors can add a more detailed literature review and some discussions about the runtime for both algorithms to the revised paper. In terms of future directions, I do believe that (1) some numerical studies that bridge the current theoretical analysis and real-world implementations would be helpful here, even for illustration purposes (is the assortment algorithm too computationally expensive to be implemented?) (2) adding a short discussion about the case when rewards are non-uniform can also be helpful, even though the authors mention that it can lead to an additional factor of $1/\kappa$, such a discussion can lead to further insights. Due to the above reasons, I will keep my original score.

---

> > > ### Author Response · Authors · 2022-08-09
> > > **Thanks for your comment!**
> > >
> > > Please find in the supplementary material a first revised version of the paper including a more detailed literature review and the runtime for both algorithms.

---

### Official Review · Reviewer_eXno · 2022-07-22

**Rating:** 5
**Confidence:** 3
**Soundness:** 3 good
**Presentation:** 3 good
**Contribution:** 2 fair

**Summary:**

This paper studied dynamic multi-product pricing and assort problems under the Multinomial Logit Model (MNL). For multi-product pricing problems, they proposed the Online Newton method for multiple product pricing algorithms with $O(d\sqrt{T}\log(T))$ regret bound. For assort problems, they proposed OFU-MNL with better dependency on the problem-dependent parameter $\kappa$.

**Questions:**

The author mentioned that optimal regret bound $(O(\sqrt{T}))$ is achievable in the case of a single product with Feature-based dynamic pricing. Does it mean that if we apply the algorithm in the paper to a single product, it will lead to a slightly worse regret bound compared to the optimal one?

Can the author comment on the running of Algorithm 1? Since it requires an inverse of a Hessian matrix at every time step, it may not be as efficient as first-order methods.

**Limitations:**

There is no potential societal impact.

**Strengths And Weaknesses:**

For dynamic pricing, the paper extends the current literature to multiple products with feature-dependent price sensitivities. I believe it is an important extension and has many applications in practice. The idea is that it can consider the problem as an online convex optimization problem and use a famous technique (Online Newton step) to deal with it. However, due to the existence of uninformative prices, it can break the ONS algorithm. Therefore, the proposed algorithm needs to introduce random shock to increase exploration.

Technically, both online newton step and random shocks have been applied to dynamic pricing under different settings as the authors pointed out in the literature, therefore I am not surprised when the authors can derive the regret bound for a new setting of multiple products. I think a very strong assumption in the paper compared to the related work is that after each round $t$, the seller can observe $q_t$, the customer probability of buying products (so that the seller can update the estimate $\gamma_t$). In Xu and Wang's work (Logarithmic Regret in Feature-based Dynamic Pricing), it seems that they only require observing indicator $y_t$ of customer buying the product or not, which is more practical.

Experiment: the experiment results in Appendix D show an advantage of the proposed algorithm compared to a similar online Newton method (ONSP) with a single product. However, the experiment result is not convincing since it is based on a stochastic setting whereas all the theoretical results in this paper focus on adversarial counterparts.

---

> ### Author Response · Authors · 2022-08-02
> **Response to Reviewer eXno**
>
> Thank you for your feedback. We would  first like to answer the general concern about the technical novelty of our work and the other concerns raised in the ‘strengths and weaknesses’ section before the response to specific questions.
>
> 1) As you correctly point out, using random price shocks has already been used in the pricing literature ([30]) and the very recent work [41] also uses an ONS-based algorithm for a pricing problem. However, we consider a different problem (multiple products, adversarial contexts, feature dependent price sensitivity) and it was not clear a priori that using simultaneously an ONS based algorithm and random price shocks would lead to an algorithm with near optimal regret. In particular, the analysis to obtain our theoretical guarantees is different from the one in [41] and [30] (see Appendix B for the proof of Lemma 2.6 and derivation of Theorem 2.4 from Lemma 2.6). Furthermore, note that [30]  considers a different problem where the contexts are drawn i.i.d. from a distribution (when the contexts are assumed adversarial, they only show a $O(T^{2/3})$ regret upper bound). On the other hand,  [41] considers a single product pricing problem without price sensitivity. Unlike in our setting, all prices are informative in the setting of [41] and an $O(d\log(T))$ regret is achievable in this case.
> We would also like to emphasize that our parameter update and the one in [41] are different (see the response to the Question 3 of reviewer rfSu for additional discussion on this point).
>
> 2) `I think a very strong assumption in the paper compared to the related work is that after each round $t$, the seller can observe $q_t$, the customer probability of buying products’.
>
> We would like to clarify this point and will modify the final version of the paper to avoid any confusion. We do not assume that the seller can observe the customers’ probability of buying each product. Similarly as in [41], we only assume that the seller observes the indicator variables $y_{t,j}$. What we assume is that the purchase probabilities have some parametric form (i.e., the purchase probabilities follow the multinomial logit model with feature-based utilities).  This assumption is similar to Assumption 1 in [41], with the difference that [41] considers more general CDF whereas we focus on the special case corresponding to the multinomial logit model. Note that when updating the parameter $\gamma_t$, we use an estimate $q_{t-1}(\gamma_{t-1}, p_{t-1})$ of the purchase probabilities, which is not the real purchase probabilities $q_{t-1}(\gamma^*, p_{t-1})$.
>
> 3) ‘The experiment result is not convincing since it is based on a stochastic setting whereas all the theoretical results in this paper focus on adversarial counterparts.’
> We believe that the fact that the regret curve scales or not with the potentially exponentially small exp-concavity parameter is due to a property of the algorithms ([41] uses the exp-concavity parameter in the descent step whereas our algorithm does not) rather than due to a property of the contexts. We therefore illustrated this point for one of the simplest possible sets of contexts (gaussian contexts). We agree that it would be useful to add experiments with more diverse contexts.
>
> Questions:
>
> 1) Thank you for this question; we will make this point clearer in the final version of the paper. The problem we consider in this paper includes feature-dependent price sensitivities and adversarial contexts. To the best of our knowledge, even in the single product setting, this problem was not solved: the algorithm proposed in [6] achieves a near-optimal $d\log(T)\sqrt{T}$ regret in the presence of feature-based price sensitivities but with iid contexts instead of adversarial contexts.
>
> 2) The two main computational steps of Algorithm 1 are to update the parameter by computing the inverse of the Hessian matrix and to project back in the set of feasible parameters relative to the norm $H_t$. The time complexity of the first step is $O(d^3)$, which is reasonable when $d$ is a small constant as in the setting considered in this paper. The projection step can be done efficiently by formulating the problem as a Quadratic Programming problem. Also, note that our current analysis would not show a $\tilde{O}(\sqrt{T})$ regret bound if we replaced the ONS update with a first-order method such as the projected stochastic gradient descent used in [20].

---

### Author Response · Authors · 2022-08-02
**Response to all reviewers**

We thank the reviewers for their valuable suggestions and feedback. We are encouraged that reviewers
found our problem setting important
and our idea to combine an ONS-based method with random price shocks interesting. In the following parts, we address more specifically the questions and concerns of each reviewer.

---

### Meta-Review · Area_Chair_cPWW · 2022-08-26

**Recommendation:** Accept
**Confidence:** Certain

**Metareview:**

In this paper the authors study the problem of dynamic multi-product pricing and assort under the Multinomial Logit model (MNL). For multi-product pricing problems, they propose the Online Newton method for multiple product pricing  with provable regret bound of O(d\sqrt(T)\logT). For the assort problems, they proposed OFU-MNL with better dependency on the problem-dependent parameter.

Overall, the authors have done a good job addressing the reviewers' concerns. While there is still lots of things to do in order to update the paper to meet the suggestions of the reviewers, I think this is a good paper and worth being published at NeurIPS, subject to the aforementioned edits.

**Award:**

No

---

### Decision · Program_Chairs · 2022-09-14

Accept